# Probiotics, synbiotics and berberine in Type 2 diabetes mellitus: A systematic review, meta-analysis, and molecular dynamics simulation study

**Md. Shadin[1,2]\*, Md. Shimul Bhuia[1,2], Mohammed Alfaifi[3], Faisal H. Altemani[4], Abdullah H. Altemani[5], Faisal Alsenani[6], Na'il Saleh[7], Muhammad Torequl Islam[1,2]**

**1** Department of Pharmacy, Gopalganj Science and Technology University, Gopalganj, Bangladesh, **2** Bioinformatics and Drug Innovation Laboratory, BioLuster Research Center Ltd., Gopalganj, Bangladesh, **3** Department of Clinical Laboratory Sciences, College of Applied Medical Sciences, King Khalid University, Abha, Saudi Arabia, **4** Department of Medical Laboratory Technology, Faculty of Applied Medical Sciences, University of Tabuk, Tabuk, Saudi Arabia, **5** Department of Family and Community Medicine, Faculty of Medicine, University of Tabuk, Tabuk, Saudi Arabia, **6** Department of Pharmaceutical Sciences, College of Pharmacy, Umm Al-Qura University, Makkah, Saudi Arabia, **7** Department of Chemistry, College of Science, United Arab Emirates University, United Arab Emirates

\* mdshadin.b.pharm@gmail.com

## Abstract

### Background

Type 2 diabetes mellitus is characterized by impaired regulation of blood glucose. Probiotics, synbiotics, and berberine (BBR) have been proposed as adjunctive interventions, but their overall effectiveness remains uncertain.

### Objective

To evaluate the effects of these interventions on glycemic control and to explore a potential molecular interaction of BBR with a key carbohydrate-digesting enzyme.

### Methods

A systematic review and meta-analysis of randomized controlled trials was conducted. Pooled effects were estimated for fasting plasma glucose (FPG) and glycated hemoglobin (HbA1c) using random-effects models. A subgroup analysis compared probiotics with placebo. An exploratory computational analysis examined the interaction of BBR with α-glucosidase.

### Results

More than 30 trials involving over 2,000 participants were included. The pooled analysis showed significant but modest reductions in FPG ($-0.71$ mmol·L$^{-1}$) and HbA1c ($-0.19\%$), with substantial between-study variability. Probiotics alone also reduced

**Data availability statement:** All relevant data are within the manuscript and its Supporting Information files.

**Funding:** This research was funded by the Deanship of Research and Graduate Studies at King Khalid University through a Large Groups Project (grant number RGP1-112-46). The grant recipient participated in manuscript preparation as a co-author but had no role in data collection, data analysis, or the decision to publish.

**Competing interests:** The authors have declared that no competing interests exist.

FPG (approximately −0.80 mmol·L$^{-1}$) and HbA1c (approximately −0.21%) compared with placebo. Computational analysis indicated weaker enzyme binding for BBR than for the reference inhibitor acarbose.

## Conclusions

Probiotics, synbiotics, and BBR provide statistically significant but clinically modest improvements in glycemic control. These findings support their use as adjunctive, rather than primary, therapeutic options and highlight the need for larger and longer trials with standardized interventions.

## Systematic review registration

https://www.crd.york.ac.uk/PROSPERO/view/CRD420251116387, identifier CRD420251116387.

## Introduction

Diabetes mellitus is a group of metabolic disorders marked by high blood sugar due to insufficient insulin production, impaired insulin action, or both, leading to acute and chronic health complications [1]. Type 2 diabetes mellitus (T2DM) represents a major and growing global public health challenge [2]. According to the International Diabetes Federation, the global prevalence of diabetes is projected to rise from approximately 537 million in 2021 to over 783 million by 2045, with T2DM accounting for more than 90% of all cases· [3]. The disease is associated with substantial morbidity and mortality, primarily due to its long-term complications such as cardiovascular disease, nephropathy, neuropathy, and retinopathy [4]. Beyond health consequences, T2DM imposes a considerable economic burden on healthcare systems and societies, contributing to reduced productivity and increased healthcare expenditures [5].

The pathophysiology of T2DM is multifactorial, characterized by insulin resistance in peripheral tissues, progressive pancreatic β-cell dysfunction, and chronic hyperglycemia [6]. In recent years, growing evidence has highlighted the role of the gut microbiota in glucose homeostasis and metabolic health [7,8]. Dysbiosis, an imbalance in the composition and function of intestinal microbiota, has been linked to systemic inflammation, impaired insulin signaling, and altered energy metabolism [9]. These findings suggest that therapeutic strategies targeting the gut microbiota may provide novel avenues for the prevention and management of T2DM.

The cornerstone of T2DM management includes lifestyle modification through diet, physical activity, and weight reduction [10]. Pharmacological therapy is often required, with metformin considered the first-line agent due to its efficacy, safety profile, and cost-effectiveness [11]. Other oral antidiabetic drugs are commonly employed to achieve optimal glycemic control. These include sulfonylureas, dipeptidyl peptidase-4 (DPP-4) inhibitors, sodium-glucose cotransporter 2 (SGLT2) inhibitors, meglitinides, and thiazolidinediones. Injectable therapies, including insulin and glucagon-like peptide-1 receptor agonists, are also widely used in clinical practice

[12]. Despite these therapeutic options, many patients fail to achieve or maintain optimal glycemic targets. Long-term use of conventional medications can be limited by adverse effects such as weight gain, gastrointestinal intolerance, or hypoglycemia, and by issues related to patient adherence and affordability. Moreover, most current standard treatments do not primarily target disturbances in gut microbiota or metabolic inflammation associated with T2DM [13]. These limitations underscore the need to explore complementary and alternative therapeutic strategies that can improve glycemic control through novel mechanisms of action.

Recent advances in microbiome research have emphasized the pivotal role of gut microbiota in the development and progression of T2DM. Dysbiosis, characterized by reduced microbial diversity and altered abundance of specific taxa, has been linked to impaired glucose metabolism, chronic low-grade inflammation, and insulin resistance [9]. Experimental studies demonstrate that modulation of gut microbiota composition can improve metabolic outcomes, highlighting the gut as a potential therapeutic target in diabetes management [14]. Probiotics, defined as live microorganisms that confer health benefits when administered in adequate amounts, and synbiotics, which combine probiotics with prebiotics, have emerged as promising interventions for restoring gut microbial balance [15]. Clinical and preclinical studies suggest that probiotics may lower fasting plasma glucose (FPG) and glycated hemoglobin (HbA1c) by improving insulin sensitivity, reducing systemic inflammation, enhancing short-chain fatty acid production, and modulating the gut–brain–liver axis [16–18]. Nevertheless, the degree of benefit reported across trials has been variable, underscoring the need for systematic evaluation of their efficacy in glycemic control.

Among the emerging therapeutic compounds that target both metabolic pathways and gut microbiota, Berberine (BBR) has shown particular promise [19]. BBR, a natural isoquinoline alkaloid extracted from plants such as *Berberis* species, has been extensively investigated for its antidiabetic potential [19]. Traditionally used in Chinese medicine, BBR has demonstrated glucose-lowering, lipid-regulating, and anti-inflammatory effects in both clinical and preclinical models [20]. Several randomized controlled trials (RCTs) report that BBR supplementation can significantly reduce FPG and HbA1c, making it a promising adjunct or alternative to conventional therapies [19]. The mechanisms underlying the antidiabetic effects of BBR are multifaceted. BBR activates the AMP-activated protein kinase (AMPK) pathway, thereby enhancing glucose uptake and improving insulin sensitivity [21]. It also functions as an α-glucosidase inhibitor, delaying carbohydrate absorption and attenuating postprandial hyperglycemia [21]. In addition, BBR has been shown to beneficially modulate gut microbiota, suggesting a shared mechanistic pathway with probiotics [22–25]. This overlap suggests potential complementary or synergistic mechanisms that warrant further investigation.

Despite promising findings, the current evidence base is characterized by heterogeneity in study design, probiotic strains, BBR formulations, and patient populations. Some RCTs report significant improvements in glycemic markers, while others show minimal or no effect. few studies have systematically synthesized clinical outcomes across probiotics, synbiotics, and BBR while integrating mechanistic insights from molecular-level analyses. To our knowledge, no previous systematic review has comprehensively evaluated these interventions within a single analytical framework that combines clinical outcomes with exploratory computational evidence.

Given these gaps, particularly the substantial heterogeneity of clinical findings and the lack of integrated clinical–mechanistic synthesis, a rigorous evaluation of available evidence is warranted. The present study aims to conduct a systematic review and meta-analysis of RCTs evaluating the efficacy of probiotics, synbiotics, and BBR, alone or in combination, on glycemic outcomes in patients with T2DM. To complement clinical findings, we integrate molecular docking and molecular dynamics simulations to explore and validate the inhibitory effects of BBR on α-glucosidase. This integrated approach aims to provide complementary clinical and mechanistic evidence that may help clarify the potential role of these interventions as adjunctive strategies in T2DM management.

## Methods

The review protocol was registered on PROSPERO (**No:** CRD420251116387).

## Search strategy and study selection

A systematic literature search was conducted in PubMed, Scopus, Cochrane Library, Web of Science, and Embase from 2015 to 2025, using a combination of MeSH terms and free-text keywords related to "Type 2 Diabetes Mellitus," "Probiotics," "Synbiotics," "Berberine," "Fasting Plasma Glucose," and "HbA1c," combined with Boolean operators (AND/OR). The search was restricted to studies published from 2015 onward to focus on contemporary randomized trials using current diagnostic criteria for T2DM and modern probiotic and BBR formulations, and to reduce methodological heterogeneity associated with older trials.

Eligible studies included randomized controlled trials (RCTs) involving adults with T2DM, evaluating probiotics, synbiotics, and/or BBR versus placebo, standard care, or no treatment, and reporting outcomes for FPG and/or HbA1c.

Trials in which these interventions were administered alongside standard antidiabetic therapy were included when the comparator group received comparable background treatment, allowing estimation of the additive effect of the intervention. Studies involving combination interventions were retained when the primary aim was to evaluate the metabolic effects of probiotics, synbiotics, or BBR.

Non-RCTs, animal studies, reviews, conference abstracts, and studies with insufficient data were excluded. Two reviewers independently screened titles, abstracts, and full texts, resolving discrepancies by consensus or a third reviewer. The study selection process followed PRISMA 2020 guidelines and is presented in the PRISMA flow diagram (Fig 1). Two authors were initially listed at registration; additional contributors participated in data extraction, analysis, and manuscript preparation and are included as co-authors in accordance with standard authorship criteria.

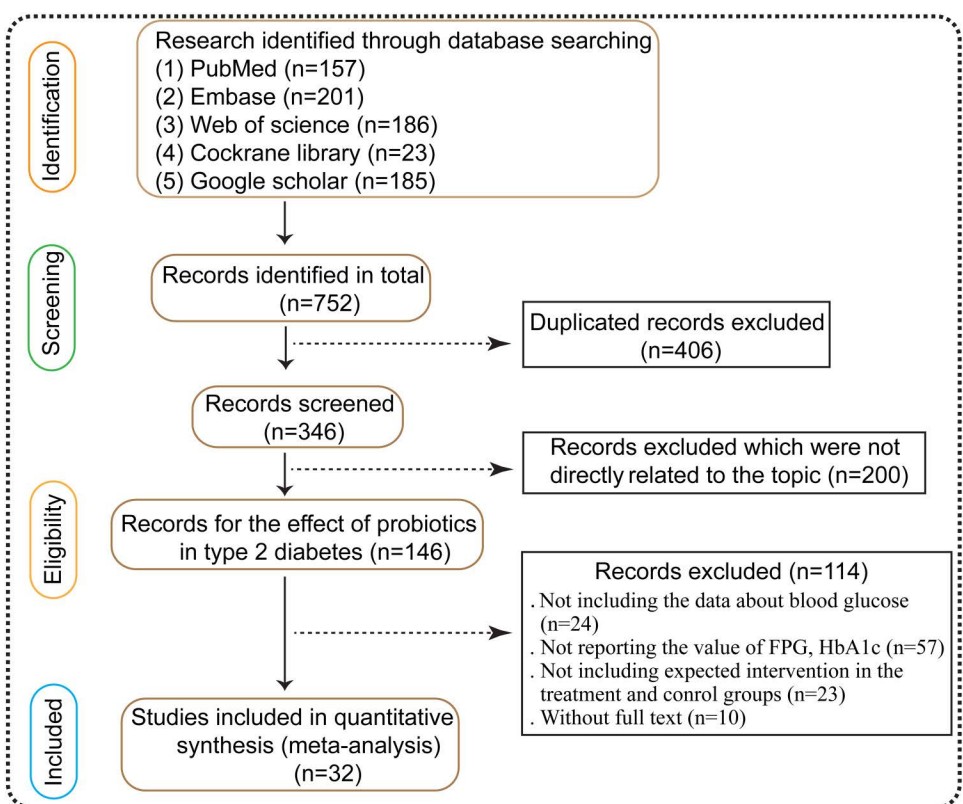

**Fig 1. The Prisma flow diagram.**

## Data extraction and quality assessment

Data from the included randomized controlled trials were independently extracted by two reviewers using a prede-signed form, capturing study characteristics (author, year, country, sample size, participant demographics, diagnostic criteria), intervention details (type, strain and dosage of probiotics or synbiotics, BBR formulation, duration), comparators, and primary outcomes FPG and HbA1c along with secondary data such as adverse events and trial duration (Table 1). Discrepancies in extraction were resolved by consensus or consultation with a third reviewer. The methodological quality of each trial was assessed using the Cochrane Risk of Bias 2.0 tool, which evaluates sequence generation, allocation concealment, blinding of participants and personnel, blinding of outcome assessment, completeness of outcome data, and selective reporting. Studies were classified as low, unclear, or high risk of bias across domains, and overall quality grading informed sensitivity analyses. This rigorous extraction and appraisal ensured reliability of pooled estimates and minimized potential biases in interpreting the clinical efficacy of probiotics, synbiotics, and BBR in T2DM.

## Data synthesis and analysis

Continuous outcomes, including FPG and HbA1c, were analyzed as mean differences (MDs) with corresponding 95% confidence intervals (CIs). When trials reported outcomes as changes from baseline, these values were extracted directly. For studies reporting only baseline and final values, the change from baseline and its standard deviation (SD) were calculated using the following formula:

$$SD_{change} = SD^2_{baseline} + SD^2_{final} - (2 * Corr * SD_{baseline} * SD_{final})$$

where *Corr* represents the correlation coefficient between baseline and final measurements. In accordance with recommendations from the Cochrane Handbook, a correlation coefficient of 0.5 was assumed when this value was not reported. For each study, the treatment effect was calculated as the difference between mean changes in the intervention and control groups:

$$MD = (\triangle Treatment) - (\triangle Control)$$

For studies reporting only post-intervention (final) values without baseline measurements, these endpoint values were directly used in the meta-analysis under the assumption of comparable baseline levels between intervention and control groups, in accordance with Cochrane Handbook guidance.

The standard error (SE) of the mean difference was computed as:

$$SE_{MD} = \sqrt{\frac{SD^2_T}{N_T} + \frac{SD^2_C}{N_C}}$$

where $SD_T$ and $SD_C$ denote the $SD_S$ of change in the treatment and control groups, and $NT_N$ and $NC_N$ represent the corresponding sample sizes. These SEs were used to derive study-specific 95% confidence intervals and weights in the meta-analysis. For studies reporting continuous data as medians with interquartile ranges or ranges, means and SDs were estimated using established conversion methods. These imputed values were included to avoid unnecessary exclusion of eligible trials and to maximize statistical power. The robustness of these assumptions was evaluated through sensitivity analyses. Statistical heterogeneity was assessed using Cochran's Q test and the I² statistic, with I² values of 25%, 50%, and 75% indicating low, moderate, and high heterogeneity, respectively. A random-effects model using the restricted maximum likelihood (REML) estimator was applied to account for between-study variability.

**Table 1. The PICOS of our study.**

| PICOS terms | Details |
|---|---|
| Population | Patients diagnosed with Type 2 Diabetes Mellitus (T2DM) were included, regardless of age, gender, or ethnicity. Diagnosis was based on ADA or WHO criteria. Studies involving patients with Type 1 diabetes, gestational diabetes, or severe comorbidities were excluded. |
| Intervention & comparators | The intervention group consisted of patients receiving probiotics or synbiotics, administered either as standalone supplements or in combination with standard care (e.g., metformin therapy, dietary management, and lifestyle modification). The comparator group included placebo, no intervention, standard care alone, or matched control products (e.g., conventional yogurt in food-based interventions). Eligible studies were randomized controlled trials that compared: (i) probiotics or synbiotics versus placebo or no intervention; (ii) probiotics or synbiotics versus standard care alone; (iii) probiotics or synbiotics combined with standard care versus standard care alone; or (iv) probiotic-enriched foods versus corresponding non-probiotic control foods. Studies in which probiotics were combined with other non-standard pharmacological or herbal agents were excluded to avoid confounding effects and to ensure that the observed outcomes could be attributed primarily to probiotics or synbiotics. |
| Outcome | Primary outcomes: – Fasting Plasma Glucose (FPG) in mmol/L – Glycated Hemoglobin (HbA1c) in % |
| Study design | Included studies met the following criteria: 1. Randomized Controlled Trials (RCTs). 2. Included at least one group receiving probiotics or synbiotics 3. Reported outcome data for FPG and/or HbA1c with means, standard deviations (SD), and sample sizes suitable for meta-analysis. 4. Provided full-text availability in any language. |
| Exclusion Criteria | Studies were excluded if they: 1. Were non-randomized, observational, or animal studies. 2. Did not report sufficient data for calculating effect sizes (e.g., missing baseline mean, mean changes, or N). 3. Involved other forms of diabetes (e.g., Type 1 or gestational). 4. Were review articles, conference abstracts, letters, or duplicate publications. 5. Used complex multi-agent interventions where the effect of probiotics could not be isolated. |

Because probiotics, synbiotics, and BBR represent distinct but clinically related adjunctive interventions aimed at improving glycemic control in T2DM, pooled analyses were performed to estimate an overall effect on glycemic outcomes while acknowledging potential mechanistic differences between interventions. Subgroup analyses were conducted according to intervention duration to explore potential sources of heterogeneity. Where possible, subgroup analyses comparing probiotics versus placebo were also performed to examine intervention-specific effects. Sensitivity analyses were performed by sequential removal of individual studies to assess the stability of pooled estimates.

Publication bias was evaluated by visual inspection of funnel plots and Egger's regression test when at least 10 studies were available for a given outcome, in accordance with meta-analysis guidelines. All statistical analyses were performed using R software (version 4.4.2) with the meta and metafor packages. Related code and data are available at: https://github.com/MDshhadinmd775/PONE-D-25-68086.git.

## The quality of evidence

The certainty of evidence for each outcome (FPG and HbA1c) was assessed using the Grading of Recommendations Assessment, Development, and Evaluation (GRADE) approach. This framework evaluates the overall quality of evidence across studies by considering risk of bias, inconsistency, indirectness, imprecision, and publication bias. Evidence was categorized as high, moderate, low, or very low certainty. Summary of Findings (SoF) tables were generated using GRADEpro GDT software to present the strength of evidence in a transparent and standardized manner.

### In silico

**Selection and preparation of receptors.** The crystal structure of human α-glucosidase (PDB ID: 3TOP, chain A) was selected as the receptor due to its high resolution and key role in glucose metabolism [26,27]. The structure was retrieved from the Protein Data Bank (https://www.rcsb.org/) and preprocessed using UCSF Chimera by removing water molecules, heteroatoms, and non-essential ligands [28]. Hydrogen atoms were added to satisfy valencies, and Gasteiger charges were assigned [29]. To further refine the geometry, energy minimization of the receptor was performed using Swiss-PdbViewer with the GROMOS96 force field, ensuring structural stability and removal of steric clashes [30]. The optimized receptor structure was then saved in PDBQT format, making it suitable for subsequent docking simulations.

**Ligand preparation.** The bioactive compound BBR was selected as the ligand due to its reported antidiabetic potential and ability to modulate glucose metabolism ([20]. The 3D structure of BBR was retrieved from the PubChem database (https://pubchem.ncbi.nlm.nih.gov/). The energy minimization was performed with the MM2 force field using Chem3D to achieve the most stable conformation [30]. The minimized structure was then saved in SDF format, followed by conversion into PDBQT format using PyRx for compatibility with molecular docking [30]. Hydrogen atoms and torsional degrees of freedom were added during the preparation step to allow conformational flexibility of the ligand in the docking simulations.

### Molecular docking

Molecular docking was performed to evaluate the binding affinity of BBR with human α-glucosidase (PDB ID: 3TOP, chain A). The docking protocol was validated by re-docking the co-crystallized ligand acarbose (ACB) into its native binding site. The active site was defined based on the ACB binding pocket and further confirmed using the CASTp server, which identifies surface pockets and cavities of proteins. Docking of BBR was conducted using PyRx with the AutoDock Vina engine. The optimized receptor and ligand structures were imported into PyRx, and the grid box was centered on the ACB binding pocket (x, y, z: −31.493, 31.987, 28.874). An exhaustiveness value of 8 was used for conformational sampling. Docking was performed in triplicate, and binding poses were evaluated based on lowest binding energy and consistency of binding orientation across runs, with clustering used to identify the most representative pose. Docking results were expressed as binding affinity values (kcal/mol), and the selected complex was further analyzed for key interactions with the active-site residues of α-glucosidase.

### Molecular dynamics simulation

Molecular dynamics (MD) simulations were carried out using GROMACS version 2025.1 (https://www.gromacs.org/) [31]. Prior to simulation, all protein ligand complexes were prepared by removing heteroatoms and resolving missing atoms or residues. The CHARMM36 all-atom force field was used for proteins, and ligand topologies were generated with CGenFF (https://app.cgenff.com/). Systems were placed in a cubic box with ≥1.5 nm padding and solvated with the CHARMM-modified TIP3P water model. Counterions ($Na^+$/$Cl^-$) were added to neutralize charge. Energy minimization used steepest descent (max 50,000 steps) with the Verlet cutoff scheme. Equilibration was performed with position restraints in NVT and NPT ensembles (100 ps each). Temperature was maintained at 300 K using the velocity-rescale thermostat; pressure at 1 bar with the Parrinello–Rahman barostat. Long-range electrostatics were treated with the particle mesh Ewald (PME) method, and Lennard–Jones interactions were computed with a force-switch cutoff scheme (switching at 1.0 nm, cutoff at 1.2 nm) without analytic dispersion correction. All bonds involving hydrogens were constrained with LINCS. Production simulations were run for 100 ns, a duration commonly used to ensure stabilization and convergence of protein–ligand complexes, which was confirmed by monitoring RMSD plateau behavior [32], using a 2 fs timestep with the leap-frog integrator; coordinates and energies were saved every 10 ps. Post-simulation analyses including RMSD (backbone), RMSF (Cα atoms), SASA, hydrogen bonds, and radius of gyration (Rg), principal component analysis (PCA) and free energy

landscape (FEL) were carried out with standard GROMACS utilities and Python scripts implemented for trajectory analysis, which are publicly available in the study GitHub repository.

## MMPBSA and energy calculation

The binding free energy of the protein ligand complex was estimated using the gmx_MMPBSA package (v1.6.4) in combination with GROMACS 2025.1 [33]. Calculations were performed using the CHARMM36 all-atom force field [34], consistent with the MD simulations. A 100 ns production trajectory was used as input, and the centered trajectory (md_0_10_center.xtc) was analyzed from frame 1–10,000, extracting one frame every 10 steps (1,000 frames total).

The Poisson–Boltzmann implicit solvent model was applied for polar solvation energy calculations, with dielectric constants of 2 (solute) and 80 (solvent) and an ionic strength of 0.15 M. Non-polar solvation energy was estimated using the SASA method. Van der Waals and electrostatic interactions were computed using the same cut-off and PME parameters as in the MD production runs. Per-residue energy decomposition was performed to identify key binding residues. Energy components (van der Waals, electrostatic, polar solvation, and non-polar solvation) and total binding free energies were obtained using the gmx_MMPBSA_ana module and are reported as mean ± SD [35].

## Pharmacokinetics and toxicity prediction

The pharmacokinetic properties of BBR were predicted using SwissADME and ProTox-III servers. SwissADME was employed to assess drug-likeness (Lipinski's rule of five), absorption, distribution, gastrointestinal permeability, blood–brain barrier penetration, and bioavailability scores [36]. Toxicity prediction was conducted using ProTox-III, which classified BBR into toxicity classes and provided toxicity estimates. These in silico analyses provided complementary insights into the pharmacokinetic behavior and safety profile of BBR as a potential therapeutic agent.

## Ethics statement

This study is a secondary analysis of previously published data and therefore did not require independent ethical approval.

## Results

### Literature search

A total of 1,248 records were identified across databases, of which 312 duplicates were removed. After title and abstract screening, 146 full-text articles were assessed for eligibility, and 32 randomized controlled trials (RCTs) met the inclusion criteria and were included in the quantitative synthesis (see PRISMA flow diagram) [34,37–66]. Full search strategy, screening details and reasons for exclusion are provided in the protocol and PRISMA flowchart, and study-level characteristics are summarised in Tables 2 and 3.

### Description of included studies

A total of 32 randomized controlled trials (RCTs) published between 2015 and 2025 were included, enrolling participants with T2DM across diverse geographical regions including Asia, Europe, North America, and the Middle East. Sample sizes varied widely, with total study enrollment ranging from 13 participants (8 intervention and 5 control) [50] to 213 participants (103 intervention and 110 control) [34], 20, with individual group sizes varying between 5 and 110 participants.. The duration of interventions ranged from 3 weeks to 104 weeks, with most trials reporting follow-up between 8 and 24 weeks. Interventions included probiotics (administered as capsules, powders, yogurts, or fermented foods), synbiotics, BBR, or their combinations, compared against placebo, conventional care, or dietary interventions. The primary outcomes consistently assessed were FPG, mmol/L; and (HbA1c, %), while several studies also reported secondary endpoints such as

**Table 2. Summary of the characteristics of studies included in the meta-analysis (<12 weeks).**

| Study | Drug | Parameters | Mean difference Treatment | N Treatment | Mean difference Control | N Control | Length of trial |
|---|---|---|---|---|---|---|---|
| [44] | Probiotic vs placebo | FPG | −0.11 | 28 | 0.11 | 25 | 10 weeks |
| | | HbA1c | −0.9 | 40 | −0.4 | 40 | 10 weeks |
| [48] | Probiotic-Smectite group vs placebo | FPG | −1.63 | 24 | −0.41 | 24 | 8 weeks |
| | | HbA1c | −0.41 | 24 | 0.26 | 24 | 8 weeks |
| [50] | Diet vs FMT | FPG | 0.3 | 8 | −2.9 | 5 | 3 weeks |
| | | HbA1c | −1 | 8 | −0.7 | 5 | 3 weeks |
| [53] | Placebo vs Probiotics | FPG | −0.78 | 22 | −0.06 | 22 | 8 weeks |
| | | HbA1c | −0.24 | 31 | −0.09 | 22 | 8 weeks |
| [54] | Placebo vs Probiotics | HbA1c | −0.67 | 22 | 0.11 | 23 | 6 weeks |
| [55] | Placebo vs Probiotics | FPG | −0.3 | 48 | 0.1 | 53 | 6 weeks |
| | | HbA1c | −0.14 | 48 | −0.01 | 53 | 6 weeks |
| [56] | Placebo vs Probiotics | FPG | −1.58 | 20 | 0.07 | 20 | 8 weeks |
| | | HbA1c | −0.24 | 20 | 0.08 | 20 | 8 weeks |
| [58] | Placebo vs Probiotics | FPG | 7.32 | 34 | 7.23 | 34 | 8 weeks |
| | | HbA1c | 6.9 | 34 | 6.9 | 34 | 8 weeks |
| [60] | Probiotic Soy Milk vs placebo | FPG | −0.06 | 20 | −0.06 | 20 | 8 weeks |
| [61] | Vitamin D + probiotic group vs Placebo | FPG | −0.35 | | −0.14 | 30 | 2 weeks |

lipid profiles, insulin resistance indices, inflammatory markers, or adverse events. The majority of probiotic interventions used multi-strain formulations, although some trials employed single strains such as *Lactobacillus* or *Bifidobacterium*. BBR was administered either alone or in combination with probiotics to explore potential synergistic effects. Despite methodological differences in sample size, strain composition, and intervention duration, the included studies provided sufficient data for quantitative synthesis of glycemic outcomes.

Across the body of evidence, certain patterns emerged. More recent studies (2020–2025), 16 and 50% of studies, increasingly focused on probiotics, often with multi-strain formulations, whereas earlier work included more single-strain or synbiotic interventions. BBR trials were predominantly conducted in Asian populations, especially in China, reflecting its traditional use and availability, while probiotic-based interventions were more geographically diverse. Longer-duration trials (≥24 weeks) tended to assess probiotics rather than BBR, suggesting greater feasibility for sustained dietary supplementation. Although the majority of studies reported improvements in FPG and HbA1c, the magnitude of benefit varied considerably by formulation, baseline glycemic status, and study quality, underscoring the importance of pooled quantitative synthesis.

## Risk of bias

Risk of bias was evaluated with the Cochrane Risk of Bias 2.0 tool across the prespecified domains. Of the 32 included randomized controlled trials, 24 trials (75%) were judged as low risk of bias, 7 trials (22%) as having some concerns, and 1 trial (3%) as high risk of bias. The most frequent limitations were incomplete reporting of the randomization process (domain D1), insufficient detail on deviations from intended interventions (D2), and limited information regarding the selection of reported results (D5). Conversely, missing outcome data (D3) and measurement of the outcome (D4) were generally rated as low risk across the majority of trials. The single high-risk study showed multiple domain concerns, notably in deviations from intended interventions, outcome measurement, and selective reporting, and should be interpreted with caution. Overall, the evidence base can be considered of moderate methodological quality: most trials contributed reliable

**Table 3. Summary of the characteristics of studies included in the meta-analysis (≥ 12 weeks).**

| Study | Drug | Parameters | Mean difference Treatment | N Treatment | Mean difference Control | N Control | Length of trial |
|---|---|---|---|---|---|---|---|
| [37] | Placebo vs WBF-011 (probiotics) | HbA1c | −0.2 | 21 | 0.4 | 16 | 12 weeks |
| | | FPG | −0.167 | 21 | 0.156 | 16 | 12 weeks |
| [67] | Placebo vs placebo+BBR | HbA1c | −1.04 | 106 | −0.59 | 103 | 12 weeks |
| [38] | Placebo vs probiotics | FPG | −0.2 | 30 | 0.1 | 30 | 12 weeks |
| | | HbA1c | −0.2 | 30 | 0.1 | 30 | 12 weeks |
| [39] | Synbiotic vs placebo | FPG | 0.34 | 42 | 0.14 | 44 | 24 weeks |
| | | HbA1c | 0.2 | 42 | 0.1 | 44 | 24 weeks |
| [40] | Placebo vs probiotics | FPG | −1.39 | 46 | −0.28 | 45 | 26 weeks |
| | | HbA1c | −0.73 | 46 | −0.14 | 45 | 26 weeks |
| [41] | Placebo vs probiotics | FPG | −0.53 | 7 | 1.28 | 7 | 12 weeks |
| | | HbA1c | 0.14 | 7 | −0.26 | 7 | 12 weeks |
| [34] | Placebo vs probiotics | FPG | −0.97 | 103 | −0.9 | 110 | 16 weeks |
| | | HbA1c | −0.44 | 103 | −0.33 | 110 | 16 weeks |
| [42] | Placebo vs probiotics | FPG | −1.4 | 40 | −0.63 | 40 | 12 weeks |
| | | HbA1c | −0.31 | 40 | 0.13 | 40 | 12 weeks |
| [43] | MET-PRO vs MET | FPG | −2 | 66 | 0.11 | 66 | 12 weeks |
| | | HbA1c | 0.9 | 66 | 0.4 | 66 | 12 weeks |
| [45] | Probiotic yogurt vs Conventional | FPG | −1.09 | 36 | −0.23 | 36 | 12 weeks |
| | | HbA1c | −0.76 | 36 | −0.15 | 36 | 12 weeks |
| [46] | Placebo vs Probiotics | FPG | 5.25 | 21 | 5.23 | 18 | 104 weeks |
| [47] | Placebo vs probiotics | HbA1c | 0.1 | 65 | 0 | 65 | 12 weeks |
| [49] | Placebo vs Probiotics | FPG | −2.87 | 42 | −1.05 | 34 | 12 weeks |
| | | HbA1c | −0.87 | 42 | −0.33 | 34 | 12 weeks |
| [51] | Berberine + Bifido-bacterium vs Placebo | FPG | −0.42 | 49 | 0.13 | 99 | 16 weeks |
| | | HbA1c | −0.22 | 49 | 0 | 99 | 16 weeks |
| [52] | Synbiotics vs Placebo | FPG-16 | 0.61 | 12 | −0.61 | 14 | 12 weeks |
| | | HbA1c | 0.27 | 12 | 0.18 | 14 | 12 weeks |
| [55] | Placebo vs Probiotics | FPG | −0.1 | 48 | 0.3 | 53 | 12 weeks |
| | | HbA1c | −0.14 | 48 | 0.02 | 53 | 12 weeks |
| [57] | ADR-1a vs Placebo | FPG | −0.02 | 22 | −0.58 | 22 | 12 weeks |
| | | HbA1c | −0.39 | 22 | 0.22 | 22 | 12 weeks |
| [58] | Placebo vs Probiotics | FPG | 7.39 | 34 | 7.62 | 34 | 16 weeks |
| | | HbA1c | 7.1 | 34 | 6.9 | 34 | 16 weeks |
| [59] | Lactobacillus reuteri DSM 17938 vs Placebo | FPG | −1.6 | 14 | 0 | 15 | 12 weeks |
| | | HbA1c | 0.1 | 14 | 0 | 15 | 12 weeks |
| [62] | UB0316 vs Placebo | FPG | −1.18 | 40 | −0.16 | 40 | 12 weeks |
| | | HbA1c | −0.5 | 40 | 0.4 | 40 | 12 weeks |
| [63] | Probiotic & selenium group vs Placebo | FPG | −0.7 | 27 | −0.11 | 27 | 12 weeks |
| [64] | Berberine vs lifestyle intervention | FPG | −2.6 | 41 | −0.7 | 39 | 16 weeks |
| [65] | Placebo vs Probiotics | FPG | −4.85 | 31 | 0.9 | 30 | 26 weeks |
| [66] | Placebo vs Probiotics | FPG | −1.3 | 30 | 0.07 | 30 | 12 weeks |
| | | HbA1c | −0.1 | 30 | −0.003 | 30 | 12 weeks |

data, but the presence of reporting deficiencies in a subset of studies and one trial at high risk of bias warrants cautious interpretation of pooled estimates. A detailed, study-level domain breakdown is presented in Fig 2.

## Main results of the meta-analysis

**Fasting plasma glucose.** The overall meta-analysis including all eligible studies demonstrated a significant reduction in FPG in the intervention group compared with control. Using a random-effects model, the pooled mean difference was −0.73 mmol/L (95% CI: −1.18 to −0.29; p = 0.0021), indicating a statistically significant but modest improvement in glycemic control (Fig 3). Substantial heterogeneity was observed across studies (I² = 87.4%, p < 0.0001), reflecting variability in study design, intervention duration, and treatment regimens. Given this heterogeneity, the pooled estimate should be interpreted as an average effect across diverse interventions rather than as the effect of a single therapeutic strategy. Nevertheless, the direction of effect consistently favored the intervention across most studies, supporting a consistent trend toward glucose lowering. These findings suggest that probiotic- and BBR-based interventions are associated with significant reductions in FPG in patients with T2DM.

Funnel plot inspection suggested asymmetry for FPG. Egger's regression test confirmed the presence of small-study effects (t = −2.69, df = 30, p = 0.0116), indicating potential publication bias. This finding suggests that effect sizes may be overestimated, particularly in smaller trials.

In the subgroup analysis comparing probiotics alone with placebo, probiotic supplementation significantly reduced FPG levels. The random-effects meta-analysis yielded a pooled mean difference of −0.80 mmol/L (95% CI: −1.28 to −0.33; p = 0.0047) in favor of probiotics, with considerable heterogeneity (I² = 87.6%) (Fig 4). These results indicate a modest but statistically significant glucose-lowering effect of probiotics alone.

Subgroup analysis by intervention duration (Fig S2 in S1 File) showed that longer treatment periods (≥12 weeks) were associated with greater reductions in FPG (MD = −0.91 mmol/L, 95% CI: −1.52 to −0.29) compared with shorter interventions (<12 weeks; MD = −0.73 mmol/L, 95% CI: −1.18 to −0.29), suggesting a potential duration-dependent effect. Despite substantial heterogeneity across studies (I² = 87%, τ² = 1.18, p < 0.001), sensitivity analyses confirmed the robustness of the overall findings.

Although statistically significant, the magnitude of FPG reduction should be interpreted cautiously in light of heterogeneity and potential small-study effects. The certainty of evidence was rated as low due to concerns regarding risk of bias and inconsistency across trials based on GRADE assessment. Visual inspection of the funnel plot indicated some asymmetry, suggesting possible publication bias (Fig S1 in S1 File), although most studies clustered around the line of no effect, with a few outliers contributing to heterogeneity. Exclusion of these outliers did not materially change the pooled effect estimate. Timeline and bubble plot analyses of FPG are presented in Fig S3 in S1 File.

## HbA1c

The meta-analysis of HbA1c included 28 RCTs comprising 2,204 participants with T2DM. The pooled analysis demonstrated a statistically significant reduction in HbA1c favoring probiotic, synbiotic, and BBR interventions compared with control (MD = −0.19%, 95% CI: −0.34 to −0.05, p = 0.0118) (Fig 5). This corresponds to a modest absolute reduction of approximately 0.2%. Substantial heterogeneity was observed among studies (I² = 86.6%, τ² = 0.1093, p < 0.0001), likely reflecting variability in study populations, intervention types (probiotics, synbiotics, and BBR), dosages, treatment durations, and baseline glycemic control. Funnel plot inspection for HbA1c did not show marked asymmetry. Egger's regression test did not indicate significant small-study effects (t = −0.83, df = 26, p = 0.4116).

The substantial heterogeneity observed across studies likely reflects differences in intervention type, probiotic strain composition, BBR dosage, treatment duration, comparator groups, and participant characteristics. To explore potential sources of heterogeneity, subgroup analyses were conducted based on intervention duration. Longer-duration interventions (≥12 weeks; n = 853 participants) showed a modest reduction in HbA1c that did not reach

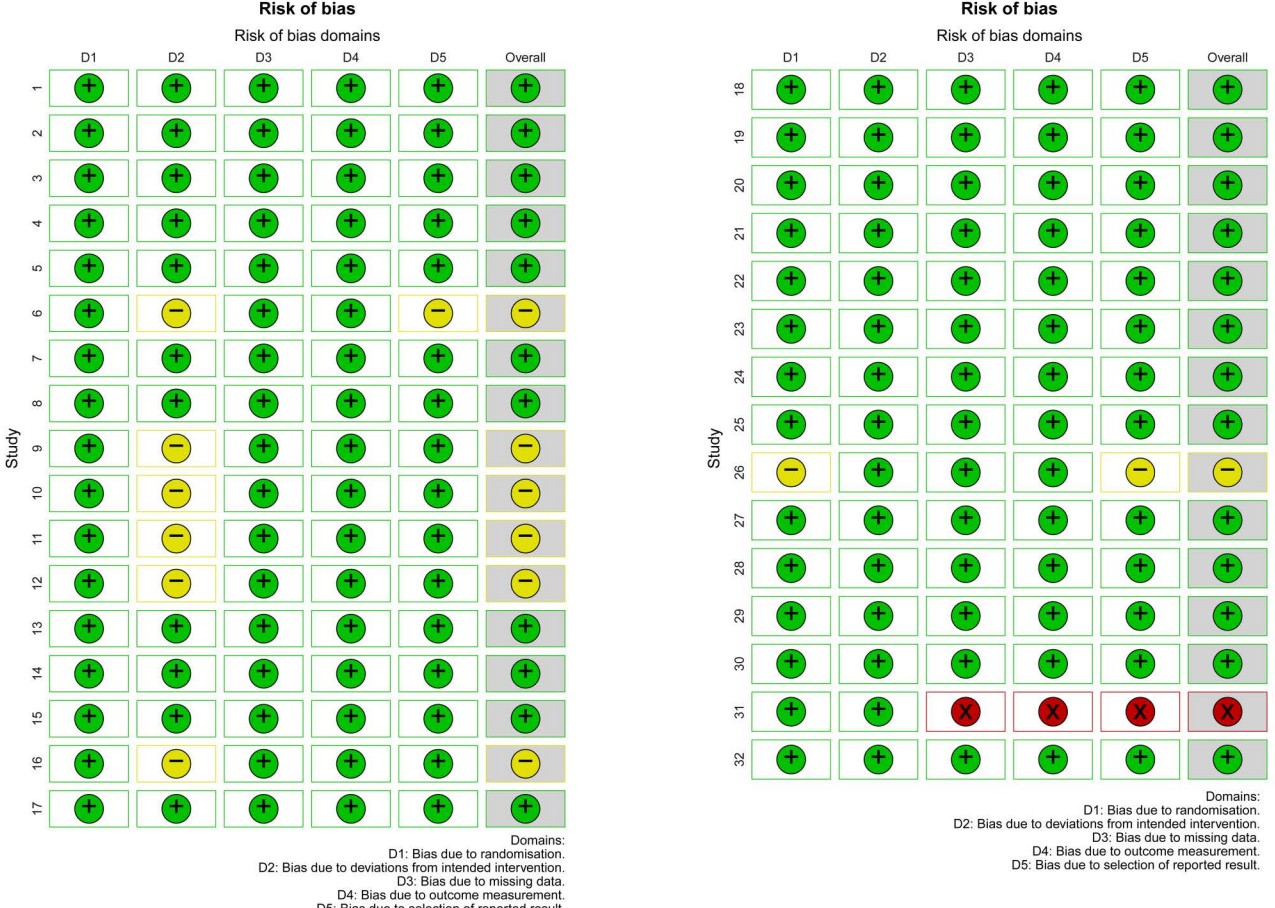

**Fig 2. The risk of bias assessment of this study.**

statistical significance (MD = −0.15%, 95% CI: −0.34 to 0.05, I² = 89.3%) (Fig S4 in S1 File). In contrast, shorter-duration interventions (<12 weeks; n = 227 participants) demonstrated a more pronounced and statistically significant reduction (MD = −0.30%, 95% CI: −0.52 to −0.09, I² = 65.8%, p = 0.0046). However, the test for subgroup differences was not statistically significant (χ² = 1.44, df = 1, p = 0.2308), indicating that intervention duration alone does not fully explain heterogeneity.

In the subgroup analysis comparing probiotics alone with placebo, probiotic supplementation significantly reduced HbA1c levels (MD = −0.21%, 95% CI: −0.37 to −0.06; p = 0.0118), with moderate-to-high heterogeneity (I² = 76.0%, p < 0.0001) (Fig 6). For this subgroup, Egger's regression test did not indicate significant small-study effects (t = −1.69, df = 8, p = 0.1297). Visual inspection of the funnel plot for HbA1c (Fig S6 in S1 File) indicated some asymmetry, suggesting possible small-study effects. Consistent with this, bubble plot analysis showed that larger studies tended to report more conservative effect estimates.

Although statistically significant, the pooled HbA1c reduction of 0.19% falls below commonly accepted thresholds for clinically meaningful change (≥0.5%). Therefore, these findings support modest adjunctive benefit rather than clinically transformative glycemic control. Timeline analysis (Fig S5 in S1 File) showed consistent HbA1c improvements across studies published between 2015 and 2025 without a clear temporal trend.

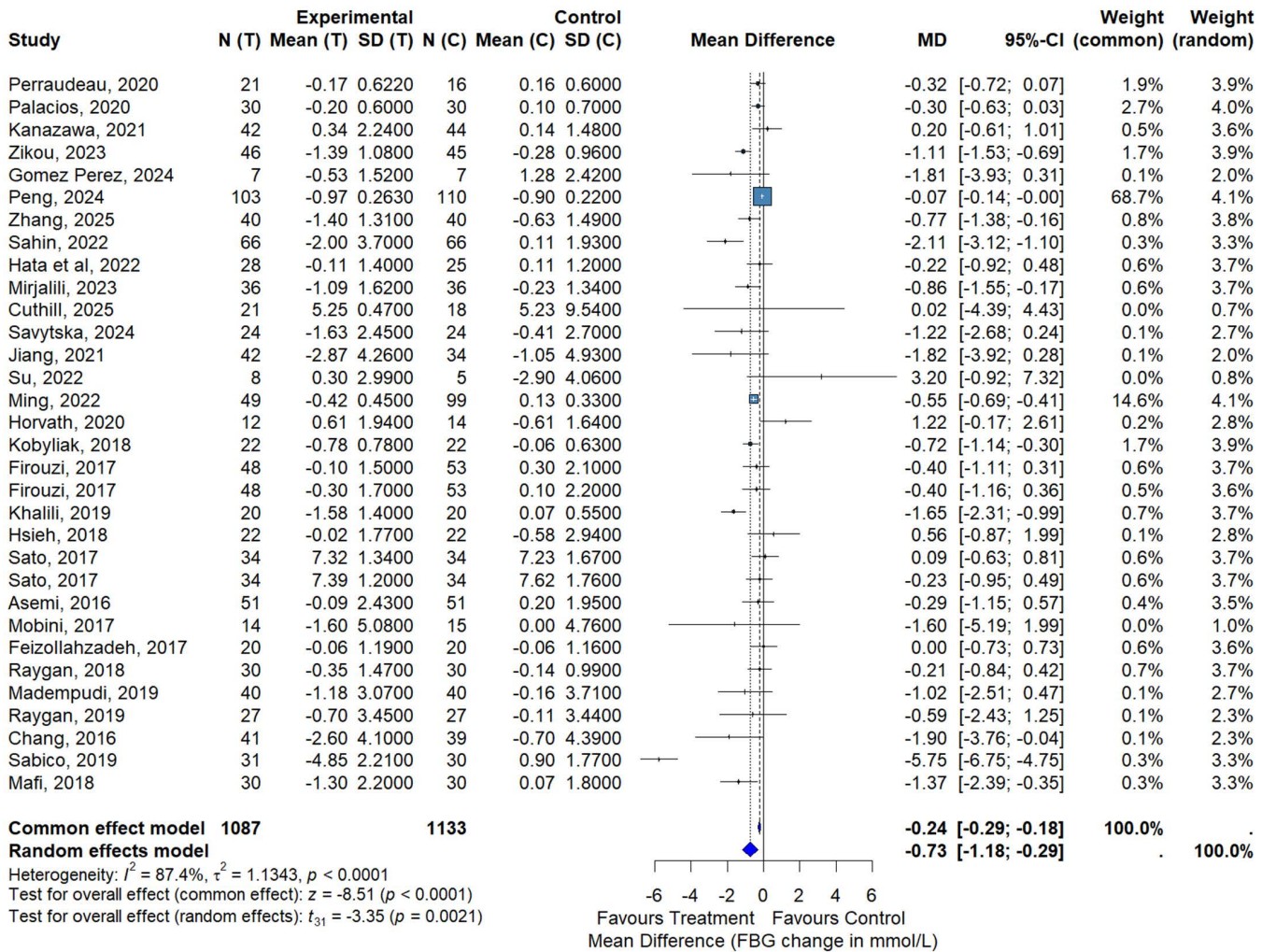

**Fig 3. Meta-analysis of the effect of probiotics, synbiotics and BBR on FPG.**

## Certainty of evidence

The overall certainty of evidence was evaluated using the GRADE framework. For FPG, the certainty was rated as low, primarily due to concerns about risk of bias in some included trials and substantial heterogeneity ($I^2 = 87\%$). Despite the consistent direction of effect favoring probiotics, synbiotics, and BBR, variability in study populations, intervention types, and durations contributed to inconsistency, leading to a downgrading of the evidence quality. For glycated haemoglobin (HbA1c), the certainty of evidence was judged to be moderate. Although most trials demonstrated reductions in HbA1c, significant between-study heterogeneity ($I^2 = 86.6\%$) warranted downgrading. Risk of bias was generally low to moderate across studies, with no serious issues related to indirectness or imprecision. Taken together, the findings indicate that probiotics, synbiotics, and BBR likely improve glycemic outcomes in patients with T2DM, but the strength of the evidence remains limited by methodological variability and inconsistency across trials. Future large-scale, high-quality RCTs with standardized interventions are required to strengthen the certainty of these results. The details are provided in Table 4.

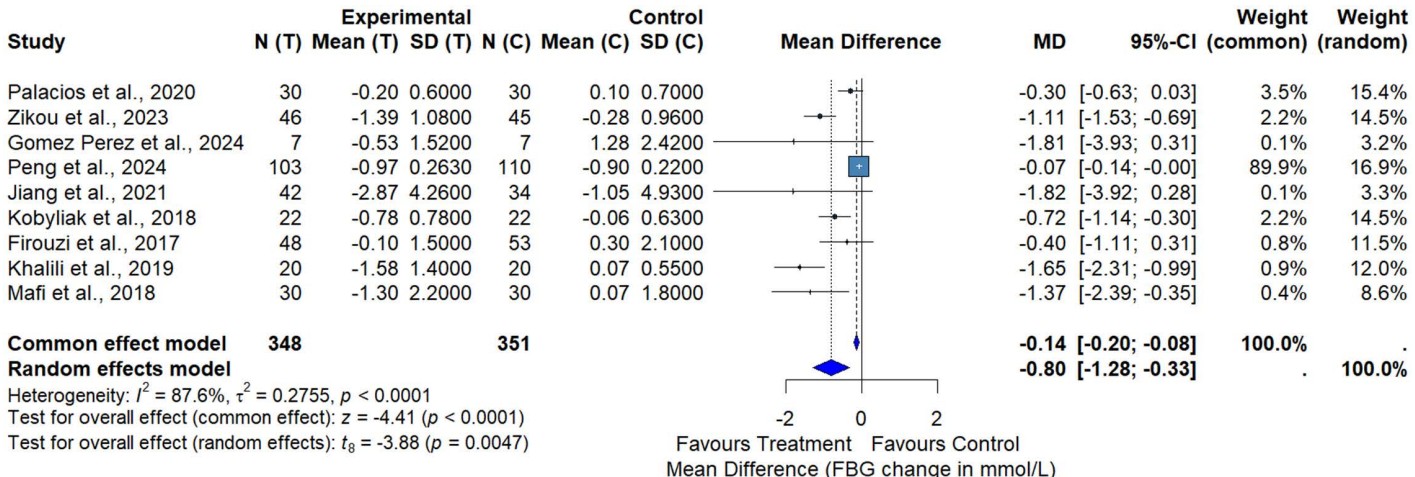

**Fig 4. Meta-analysis of the effect of probiotics versus placebo on FPG.**

## In silico

**Molecular docking.** Docking of BBR and the reference ligand ACB into the active site of human α-glucosidase (PDB ID 3TOP, chain A) was performed using AutoDock Vina as described in Methods. The best scoring pose for ACB returned a binding affinity of −8.7 kcal·mol⁻¹, forming five hydrogen bonds with ASP1157 (2.56 Å), MET1421 (3.50 Å), ASP1526 (2.92 Å), HIS1584 (3.33 Å) and ASP1279 (2.88 Å), and additional π/alkyl and carbon–hydrogen contacts with ASP1279, TYR1251, PHE1559 and PHE1560. BBR docked with a top score of −8.1 kcal·mol⁻¹, forming a single hydrogen bond to ARG1510 (≈3.04 Å) and multiple hydrophobic/aromatic contacts with TYR1251, TRP1355, PHE1559, PHE1560, PRO1159 and MET1421. These residues MET1421, ASP1526, TYR1251, PHE1559, and PHE1560 are same in both ligands and ensure the active site (Table 5 and Fig S7 in S1 File). The small difference in Vina scores (0.6 kcal·mol⁻¹) should be interpreted cautiously; ACB's greater number of polar contacts correlates with its more favorable MM-PBSA binding free energy reported below, whereas BBR appears to rely more on hydrophobic stabilization within the pocket. In the S2 File, we showed the details about MD simulation, MM/PBSA binding free energy, PCA analysis, pharmacokinetics and toxicity prediction.

## Safety and tolerability

Across the included trials, probiotics and synbiotics were consistently well tolerated, with no serious adverse events reported [68]. However, adverse event reporting was not uniform across all studies, and underreporting cannot be excluded. BBR demonstrated a generally acceptable safety profile, consistent with the findings of [19], who reported significant glucose-lowering efficacy accompanied mainly by mild gastrointestinal adverse effects such as constipation, diarrhoea, and abdominal discomfort, with serious events being rare [69]. Complementing the clinical evidence, our *in silico* analyses predicted favorable drug-likeness and oral absorption for BBR, but also indicated potential CYP-mediated drug–drug interactions and possible risks of neurotoxicity, immunotoxicity, and carcinogenicity. These computational findings are exploratory and should be interpreted with caution, as they require validation in preclinical and clinical safety studies. Taken together, probiotics appear safe and well tolerated in the short term, while BBR is generally safe in short-term use but warrants careful monitoring, particularly when used concomitantly with other medications.

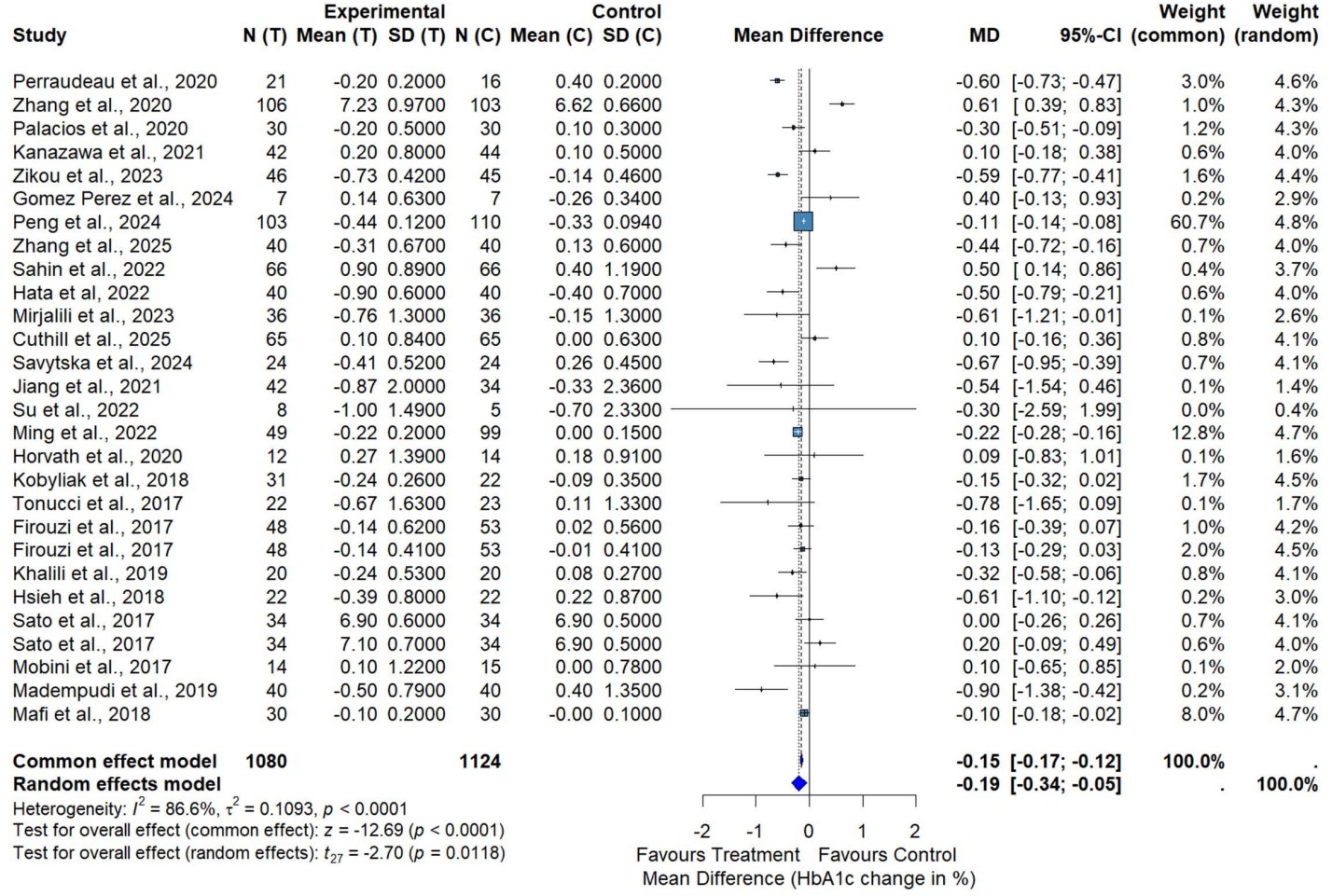

**Fig 5. Meta-analysis of the effect of probiotics, synbiotics and BBR on HbA1c.**

## Discussion

In this integrated systematic review and meta-analysis, complemented by exploratory in silico analyses, we observed statistically significant but modest improvements in glycemic control associated with probiotics, synbiotics, and BBR in patients with T2DM. The pooled mean reduction in FPG was −0.73 mmol·L$^{-1}$ and the pooled reduction in HbA1c was −0.19%. These effects were directionally consistent across most studies but were accompanied by substantial between-study heterogeneity [I²>85% for both FPG and HbA1c], which necessitates cautious interpretation. Because the included interventions (probiotics, synbiotics, and BBR) differ in their biological mechanisms but share the common therapeutic objective of improving glycemic control in T2DM, the pooled estimates should be interpreted as overall adjunctive effects across related but mechanistically distinct interventions rather than effects attributable to a single therapeutic strategy.

Although clinical and computational approaches were combined in this study, these components address distinct levels of biological inference. The molecular docking and molecular dynamics analyses were designed as exploratory investigations focused specifically on BBR–α-glucosidase interactions and do not capture the dominant mechanisms by which probiotics influence glycemic regulation [70]. Probiotics are known to exert metabolic effects primarily through modulation of gut microbiota composition, short-chain fatty acid production, bile acid metabolism, and inflammatory signaling pathways

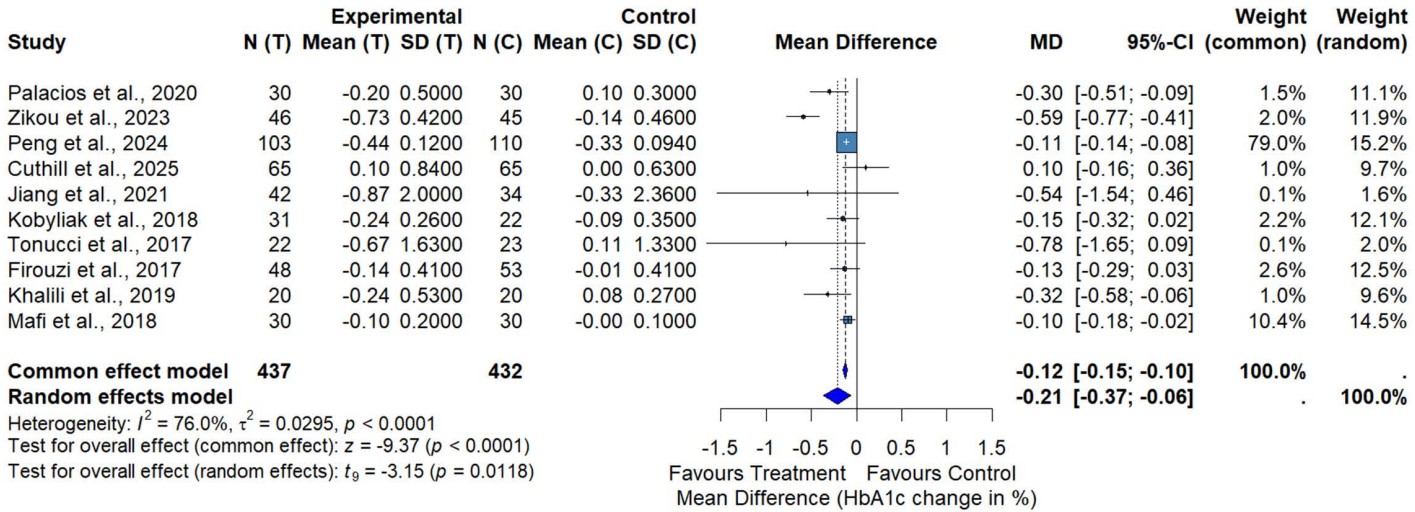

**Fig 6. Meta-analysis of the effect of probiotics versus placebo on HbA1c.**

**Table 4. Summary of findings table.**

| Outcome | Study design | Risk of bias | Inconsistency | Indirect-ness | No. of participants (studies) | Effect (MD, 95% CI) | Certainty (GRADE) & Comments |
|---------|--------------|--------------|---------------|---------------|-------------------------------|---------------------|------------------------------|
| FPG (mmol/L) | Randomized trials | Low to some concerns (most studies low risk; 1 study high risk) | Serious (downgraded) – I²=87% | Not serious | 2265 (33 RCTs) | MD=−0.71 mmol/L [95% CI: −1.15 to −0.27]; I²=87% | ●●○○○ Low – Downgraded for risk of bias (some concerns) and inconsistency (heterogeneity) |
| HbA1c (%) | Randomized trials | Low to some concerns | Serious (downgraded) – I²=86.6% | Not serious | 2204 (28 RCTs) | MD=−0.19% [95% CI: −0.34 to −0.05]; I²=86.6% | ●●●○ Moderate – Down-graded for inconsistency (heterogeneity) |

**Table 5. Docking results of berberine and acarbose against α-glucosidase, including binding affinities, number of hydrogen bonds, interacting residues, and interaction types.**

| Receptor (PDB ID) | Ligands | Binding affinity (Kcal/mol) | No of HB | Amino acid residues | |
|-------------------|---------|------------------------------|----------|---------------------|--|
| | | | | **HB (Length) (A˚)** | **Other bonds** |
| 3TOP (A chain) | BBR | −8.1 | 1 | ARG1510 (3.03727) | PHE1560 (Electrostatic), TYR1251, TRP1355, PHE1559, ASP1526, PRO1159, and MET1421 (Hydrophobic) |
| | ACB | −8.7 | 5 | ASP1157 (2.557), MET1421 (3.497), ASP1526 (2.919), HIS1584 (3.331), ASP1279 (2.881) | PHE1560 (Carbon Hydrogen Bond), ASP1279, TYR1251, PHE1559, and ASP1420 (Pi-Alkyl) |

BBR: Berberine, ACB: Acarbose, HB: Hydrogen bond.

rather than through direct inhibition of carbohydrate-digesting enzymes [71]. Similarly, important mechanisms attributed to BBR, such as AMPK activation and microbiota-mediated effects, are not represented by the chosen computational target (Fig 7) [24,72–74]. Accordingly, the *in silico* findings should be interpreted as complementary and BBR-specific and not as mechanistic validation of the pooled probiotic effects observed in the clinical meta-analysis. Thus, the computational analyses should be interpreted as exploratory mechanistic insights specific to BBR rather than as explanations for the broader pooled clinical outcomes observed across probiotics, synbiotics, and BBR interventions. Importantly, these computational

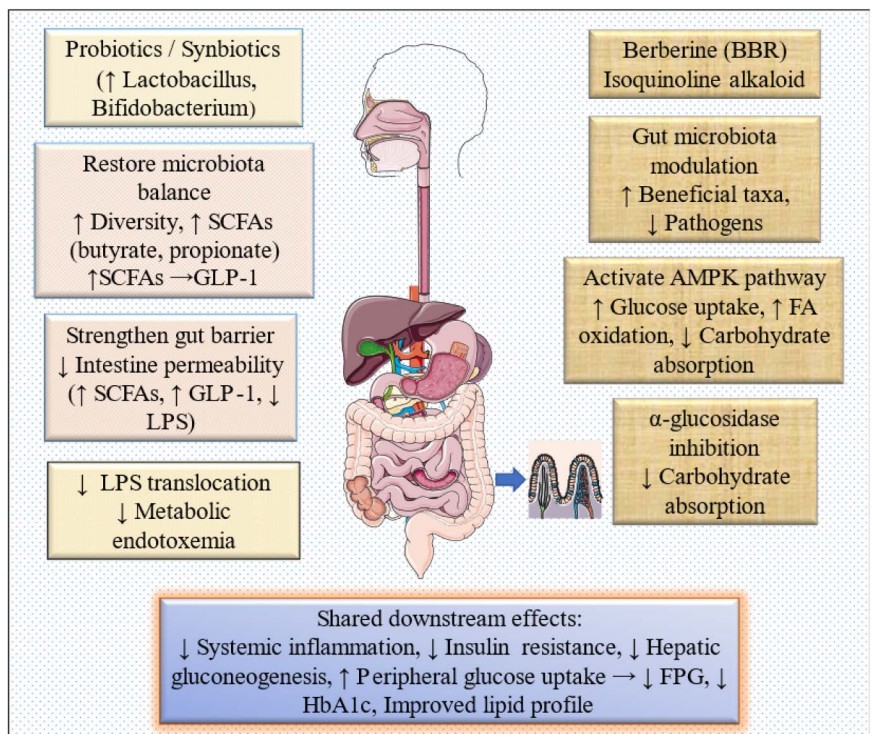

**Fig 7. Proposed mechanism: probiotics and berberine in type 2 diabetes mellitus.**

findings should not be interpreted as direct mechanistic explanations for the pooled clinical effects, but rather as hypothesis-generating insights limited to a specific molecular interaction involving BBR.

The pooled estimates for both FPG and HbA1c were characterized by very high heterogeneity ($I^2 > 85\%$), reflecting marked diversity in intervention type, strain composition, dosing regimens, comparator arms, and study populations. The included trials encompassed single-strain probiotics, multi-strain formulations, synbiotics, BBR monotherapy, and combination therapies, which differ fundamentally in biological mechanisms and clinical context. Therefore, the pooled effects should be interpreted as average adjunctive responses across heterogeneous interventions rather than as effects attributable to a single therapeutic strategy. Although subgroup analyses separating probiotics versus placebo and analyses stratified by intervention duration were performed, residual heterogeneity remained, indicating that treatment duration alone does not fully explain variability in outcomes.

In analyses restricted to probiotics versus placebo, probiotic supplementation alone produced modest reductions in FPG (approximately −0.80 mmol·L⁻¹) and HbA1c (approximately −0.21%), supporting a small but consistent adjunctive effect independent of BBR. However, the magnitude of HbA1c reduction observed in both overall and subgroup analyses falls below commonly accepted thresholds for clinically meaningful change (≥0.5%). The observed FPG reduction lies at the lower end of ranges commonly considered clinically relevant (≈0.5–1.0 mmol·L⁻¹), and therefore should be interpreted as modest and context-dependent rather than definitively clinically meaningful. These findings therefore suggest adjunctive benefits rather than effects likely to alter standard therapeutic decision-making. Therefore, any potential clinical application of these interventions should be considered within the context of individualized patient management and existing diabetes treatment guidelines.

The *in silico* analyses demonstrated that BBR interacts with α-glucosidase with weaker binding affinity and lower thermodynamic stability than the reference inhibitor ACB [75]. BBR relied predominantly on hydrophobic interactions and

formed fewer persistent hydrogen bonds within the catalytic pocket, whereas ACB occupied a deeper and more stable free energy basin. These computational observations are not mechanistic explanations of the clinical effects but represent indirect and hypothetical associations between predicted binding behavior and observed glycemic outcomes. The computational component is therefore presented as exploratory and hypothesis-generating rather than as mechanistic confirmation of clinical efficacy. Accordingly, these findings should be interpreted as supportive molecular-level observations specific to BBR, and not as explanatory of the broader clinical effects observed across heterogeneous interventions included in the meta-analysis.

Safety outcomes were generally favorable. Probiotics and synbiotics were consistently well tolerated, with no serious adverse events reported [76,77]. BBR demonstrated an acceptable short-term clinical safety profile, with predominantly mild gastrointestinal adverse effects. In silico pharmacokinetic predictions suggested favorable oral absorption but also indicated potential CYP-mediated drug–drug interactions and predicted toxicity risks [78,79]. These computational safety signals are exploratory and require experimental and clinical validation but highlight the importance of monitoring when BBR is used in combination with other medications.

Assessment of publication bias revealed evidence of small-study effects for FPG (Egger's test p = 0.0116), whereas no significant asymmetry was detected for HbA1c in the overall analysis (p = 0.4116) or in the probiotics versus placebo subgroup (p = 0.1297) [80]. Nevertheless, many included trials were small, short in duration, and likely underpowered, which may inflate effect estimates and contribute to heterogeneity. These limitations reduce confidence in the precision and generalizability of the pooled findings and underscore the need for cautious interpretation of effect sizes.

Overall, heterogeneity likely arises from differences in probiotic strain composition, BBR dose and formulation, intervention duration, baseline glycemic control, concomitant medications, and population characteristics. Methodological limitations, including incomplete reporting of randomization procedures and selective outcome reporting in some trials, further constrain interpretation. In addition, the in silico simulations model only a single molecular target and cannot capture systemic pharmacodynamics or microbiota-mediated effects that likely contribute to clinical efficacy. Additional contributors to heterogeneity may include variation in baseline glycemic control, geographic populations, background antidiabetic therapy, and differences in probiotic strain composition and formulation, which were not consistently standardized across trials.

Taken together, these findings support a potential role for probiotics, synbiotics, and BBR as adjunctive interventions associated with modest improvements in glycemic control. However, the certainty of evidence was rated as low to moderate according to GRADE, and given the limited magnitude of HbA1c reduction and the substantial heterogeneity across trials, these agents should not replace first-line pharmacotherapy in patients requiring substantial glycemic lowering. Future randomized controlled trials should be larger, longer in duration (≥24 weeks), standardized with respect to probiotic strain composition and BBR dosing, and incorporate mechanistic and safety endpoints to strengthen clinical applicability and interpretability.

## Conclusion

This integrated meta-analysis and in silico investigation suggests that probiotics, synbiotics, and BBR are associated with statistically significant but modest improvements in glycemic control in T2DM (overall FPG ≈ −0.73 mmol·L⁻¹; overall HbA1c ≈ −0.19%). In analyses restricted to probiotics versus placebo, probiotic supplementation alone also produced modest reductions in FPG (≈ −0.80 mmol·L⁻¹) and HbA1c (≈ −0.21%). Substantial heterogeneity and methodological limitations reduce confidence in the precision of these estimates. Computational analyses suggest that BBR interacts with α-glucosidase less stably than ACB, consistent with its limited clinical effect. However, these computational findings are limited to berberine-specific molecular interactions and should not be interpreted as direct mechanistic explanations for the overall pooled clinical outcomes. *In silico* pharmacokinetic predictions suggest favorable absorption but potential CYP-mediated interactions and toxicity signals that require experimental validation. Overall, these interventions may be

considered adjunctive strategies rather than primary therapies, and any potential clinical use should be guided by individualized patient assessment and established diabetes management guidelines without direct inference from the *in silico* findings. Well-powered, standardized randomized controlled trials with integrated mechanistic and safety assessments are required to define their clinical role more clearly.

## Supporting information

**S1 File. Supplementary figures related to meta-analysis and molecular docking.** This file includes funnel plots, subgroup analyses, timeline and bubble plots, and molecular docking visualizations (**Fig S1–S7**).
(DOCX)

**S2 File. Supplementary results related to molecular dynamics simulation, MM/PBSA analysis, and in silico pharmacokinetics and toxicity prediction.** This file includes RMSD, RMSF, radius of gyration, SASA, hydrogen bond analysis, binding free energy calculations, and pharmacokinetic/toxicity predictions.
(DOCX)

**S3 File. PRISMA 2020 checklist.** Checklist of items included in this systematic review and meta-analysis in accordance with PRISMA 2020 reporting guidelines.
(DOCX)

## Author contributions

**Conceptualization:** Md. Shadin, Mohammed Alfaifi, Faisal Alsenani.

**Data curation:** Md. Shadin.

**Formal analysis:** Md. Shadin.

**Investigation:** Md. Shimul Bhuia, Mohammed Alfaifi, Faisal H. Altemani, Abdullah H. Altemani, Faisal Alsenani, Na'il Saleh, Muhammad Torequl Islam.

**Methodology:** Md. Shadin.

**Project administration:** Md. Shadin.

**Resources:** Md. Shadin.

**Software:** Md. Shadin.

**Supervision:** Mohammed Alfaifi, Faisal H. Altemani, Abdullah H. Altemani, Faisal Alsenani, Na'il Saleh, Muhammad Torequl Islam.

**Validation:** Md. Shimul Bhuia, Mohammed Alfaifi, Faisal H. Altemani, Abdullah H. Altemani, Faisal Alsenani, Na'il Saleh, Muhammad Torequl Islam.

**Visualization:** Md. Shadin.

**Writing – original draft:** Md. Shadin.

**Writing – review & editing:** Md. Shadin.

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
