## [Decision Letter · Decision Letter 0]

23 Feb 2026

PONE-D-25-68086Probiotics and Berberine in Type 2 Diabetes Mellitus: A Systematic Review, Meta-Analysis, and Molecular Dynamics Simulation StudyPLOS One

Dear Dr. Shadin,

Thank you for submitting your manuscript to PLOS ONE. After careful consideration, we feel that it has merit but does not fully meet PLOS ONE’s publication criteria as it currently stands. Therefore, we invite you to submit a revised version of the manuscript that addresses the points raised during the review process.

We look forward to receiving your revised manuscript.

Kind regards,

Yusuf Oloruntoyin Ayipo, Ph.D

Academic Editor

PLOS One

Journal Requirements:

“This research did not receive any specific grant from funding agencies in the public, commercial, or not-for-profit sectors.”

5. Please ensure that you refer to Figures 11, 12 and 15 in your text as, if accepted, production will need this reference to link the reader to the figure.

6. We note you have included a table to which you do not refer in the text of your manuscript. Please ensure that you refer to Tables 1 and 4 in your text; if accepted, production will need this reference to link the reader to the Table.

Additional Editor Comments:

The submission reflects scientific relevance. However, some fundamental issues limit its quality for publication in the current form. For instance, the authors need to justify the significance of the study, relate it to the literature and identify the gap in the existing knowledge that this study aims to address, and ensure an adequate validation of the theoretical analyses reported. Again, what are the limitations of this study, and how can the authors recommend future research on the study? Moreover, some serious concerns have been raised by the reviewers affecting pivotal sections of the study. Kindly pay close attention to these and address them critically before resubmission.

Reviewers' comments:

Reviewer's Responses to Questions

**Comments to the Author**

1. Is the manuscript technically sound, and do the data support the conclusions?

Reviewer #1: Yes

Reviewer #2: Yes

Reviewer #3: Partly

2. Has the statistical analysis been performed appropriately and rigorously? 

Reviewer #1: Yes

Reviewer #2: Yes

Reviewer #3: No

3. Have the authors made all data underlying the findings in their manuscript fully available?

Reviewer #1: Yes

Reviewer #2: Yes

Reviewer #3: Yes

4. Is the manuscript presented in an intelligible fashion and written in standard English?

Reviewer #1: Yes

Reviewer #2: Yes

Reviewer #3: Yes

5. Review Comments to the Author

Reviewer #1: This manuscript addresses an important and timely topic: the potential role of probiotics, synbiotics, and berberine as adjunctive interventions in the management of type 2 diabetes mellitus. The inclusion of a large number of randomized controlled trials, adherence to PRISMA guidelines, PROSPERO registration, and the use of GRADE to assess certainty of evidence are strengths of the study. The topic is relevant to both clinical and translational research audiences, and the authors have made a substantial effort to synthesize a broad and complex literature.

However, several major issues limit the interpretability and robustness of the findings and should be addressed before the manuscript can be considered for publication.

First, there is a conceptual mismatch between the clinical meta-analysis and the in-silico component of the study. The molecular docking and molecular dynamics analyses focus exclusively on α-glucosidase inhibition, which does not reflect the dominant biological mechanisms by which probiotics influence glycemic control. In addition, key mechanisms attributed to berberine, including AMPK activation, modulation of gut microbiota, and bile acid signaling, are not captured by the chosen computational target. As a result, the computational analyses do not adequately explain the pooled clinical outcomes, particularly for probiotic interventions, and risk giving an impression of mechanistic validation that is not fully supported. The authors should either clearly reframe the in-silico work as exploratory and berberine-specific or remove strong claims suggesting consistency between the computational findings and the overall clinical results.

Second, the meta-analysis pools interventions with substantial biological and clinical heterogeneity. The included studies encompass single-strain probiotics, multi-strain probiotics, synbiotics, berberine monotherapy, combination therapies, and diet or fecal microbiota transplantation comparators. These interventions differ fundamentally in mechanisms of action and clinical context. The very high heterogeneity observed for both fasting plasma glucose and HbA1c (I² > 85%) suggests that the pooled estimates may represent clinically uninterpretable averages. Subgroup analysis based only on treatment duration is insufficient to address this issue. Separate meta-analyses for probiotics/synbiotics alone, berberine alone, and combination therapies (where data permit) would substantially improve interpretability and analytical rigor.

Third, the clinical significance of the findings appears overstated. The pooled reduction in HbA1c of approximately −0.19% is below commonly accepted thresholds for clinically meaningful change. While this limitation is briefly acknowledged, it is not sufficiently emphasized in the interpretation. Statements implying clinically meaningful improvement, particularly in relation to fasting plasma glucose, should be tempered. The results are more consistent with modest, adjunctive effects rather than effects likely to influence standard therapeutic decision-making.

Fourth, risk of bias and publication bias are not explored in sufficient depth. Funnel plot asymmetry is visible for both primary outcomes, suggesting potential small-study effects, yet Egger’s regression results are not quantitatively reported. Many included trials are small, short in duration, and likely underpowered, which may inflate effect sizes and contribute to heterogeneity. These issues should be explicitly addressed, with reporting of Egger’s test p-values and a clearer discussion of how potential publication bias and small-study effects may influence the conclusions.

Fifth, the in-silico section is disproportionately long relative to its contribution to the overall conclusions. Detailed reporting of molecular dynamics simulations, principal component analysis, free energy landscapes, and MM-PBSA calculations exceeds what is necessary to support the study’s claims. Given PLOS ONE’s emphasis on methodological soundness rather than mechanistic novelty, the computational results should be substantially condensed, with extended analyses and figures moved to the Supplementary Information.

There are also several minor issues that require attention. The manuscript would benefit from language editing to address run-on sentences, repetition, and inconsistent tense usage. Table 2 is extremely dense and difficult to interpret and should be simplified or divided to improve readability. Some references are dated or repeated without sufficient clarification. The ethics statement marked as “N/A” is acceptable; however, it should explicitly state that the study is a secondary analysis of previously published data.

In summary, while the study addresses a relevant question and is methodologically grounded in several respects, the issues outlined above limit confidence in the conclusions. Addressing the conceptual alignment between clinical and computational components, appropriately handling heterogeneity, tempering claims of clinical significance, and strengthening bias assessment are necessary for the manuscript to meet the standards expected for publication.

Reviewer #2: This manuscript presents a well-structured systematic review and meta-analysis of probiotics and synbiotics in type 2 diabetes, complemented by in silico mechanistic analyses of berberine against α-glucosidase. The methodology is generally sound, including appropriate meta-analytic techniques, heterogeneity assessment, and risk-of-bias evaluation, and the integration of computational modeling adds conceptual depth. However, several conclusions particularly regarding clinical relevance and mechanistic implications are occasionally overstated and should be tempered to reflect the modest effect sizes and substantial heterogeneity observed. With careful revision the manuscript would make a valuable contribution to the field.

Introduction

1. “Type 2 diabetes mellitus (T2DM) is one of the most pressing public health challenges worldwide.”

This statement is clear but somewhat generic. Add a supporting citation directly within this sentence to strengthen its impact and align with PLOS ONE’s preference for evidence-supported claims.

2.“The disease is associated with substantial morbidity and mortality, primarily due to its long-term complications such as cardiovascular disease, nephropathy, neuropathy, and retinopathy (2007).”

The citation is incomplete and does not follow proper referencing format. Replace “(2007)” with a complete and correctly formatted citation according to PLOS ONE guidelines.

3.“The pathophysiology of T2DM is multifactorial, characterized by insulin resistance in peripheral tissues, progressive pancreatic β-cell dysfunction, and chronic hyperglycemia(Galicia-Garcia et al., 2020).”

There is a spacing error before the citation. Insert a space between “hyperglycemia” and the citation. Additionally, ensure the citation format complies with PLOS ONE numbered style.

4.“Dysbiosis an imbalance in the composition and function of intestinal microbiota has been linked to systemic inflammation…”

A comma is required after “Dysbiosis.” Revise to: “Dysbiosis, an imbalance in the composition and function of intestinal microbiota, has been linked…”

5.“Furthermore, none of the current standard treatments directly address the underlying disturbances in gut microbiota or the metabolic inflammation associated with T2DM (Caturano et al., 2024).”

This statement may be overly strong, as some treatments (e.g., metformin) have documented microbiota effects. Consider softening the claim to: “Most current standard treatments do not primarily target disturbances in gut microbiota…” and ensure appropriate citation support.

6. “Recent advances in microbiome research have emphasized the pivotal role of gut microbiota in the development and progression of T2DM.”

This sentence overlaps conceptually with earlier microbiota discussion. Consider merging this paragraph with the earlier microbiota section to reduce redundancy and improve flow.

7. “Among the emerging therapeutic compounds that target both metabolic pathways and gut microbiota, BBR has shown particular promise.”

The abbreviation “BBR” is used before being defined. Revise to: “Berberine (BBR) has shown particular promise…” at first mention.

8.“In addition, BBR has been shown to beneficially modulate gut microbiota, suggesting a shared mechanistic pathway with probiotics (Huang et al., 2022; Wang et al., 2022; Wang et al., 2022; Yang et al., 2023).”

The citation “Wang et al., 2022” appears twice. Remove the duplicate reference and verify citation accuracy.

9.“This overlap raises the possibility of synergistic effects when probiotics and berberine are used in combination.”

This statement is speculative. Consider softening it to: “This overlap suggests potential complementary or synergistic mechanisms that warrant further investigation.”

10.“Critically, no previous systematic review has comprehensively evaluated the combined effects of probiotics and berberine…”

This is a strong novelty claim and may be difficult to substantiate. Revise to: “To our knowledge, no previous systematic review has comprehensively evaluated…” or “Few studies have systematically evaluated…” unless a documented search strategy confirms the claim.

11.“By combining evidence from both clinical trials and in silico analyses, our study seeks to provide robust and mechanistically informed insights…”

The wording is slightly promotional. Consider using more neutral language consistent with PLOS ONE standards, such as: “This integrated approach aims to provide complementary clinical and mechanistic evidence…”

12.“Given these gaps in the current literature, a rigorous synthesis of available evidence is warranted.”

This sentence is appropriate but could be strengthened by briefly specifying the identified gap (e.g., heterogeneity, lack of combined evaluation, absence of mechanistic validation) to improve precision.

Method

2.1. Search strategy and study selection

1.“A systematic literature search was conducted in PubMed, Scopus, Cochrane Library, Web of Science, and Embase from 2015 to 2025…”

The search timeframe restriction (2015–2025) requires justification. Briefly explain why studies published before 2015 were excluded to reduce concerns about potential selection bias.

2. “Two authors were initially listed at registration; additional authors contributed during the review and are included in this publication.”

This statement raises authorship transparency concerns. Clarify whether the PROSPERO record was formally updated and confirm that all listed authors meet established authorship criteria.

2.3. Data synthesis and analysis

1.“A random-effects model was applied to account for between-study variability.”

This statement is appropriate but incomplete. Specify which estimator was used (e.g., DerSimonian–Laird, REML) to improve methodological transparency.

2.“Publication bias was evaluated using funnel plot inspection and Egger’s test.”

Clarify whether Egger’s test was performed only when at least 10 studies were available, as recommended in meta-analysis guidelines.

3.“All statistical analyses and plots were conducted using R software.”

This statement lacks reproducibility detail. Specify the R version and the packages used (e.g., meta, metafor) to enhance transparency.

2.4. The quality of evidence

“Summary of Findings (SoF) tables were generated…”

Indicate whether GRADEpro software or another tool was used to construct the SoF tables, as this improves reproducibility.

2.5.3. Molecular docking

1.“Default exhaustiveness parameters were applied…”

The term “default” is insufficient for reproducibility. Specify the exact exhaustiveness value used in AutoDock Vina.

2. “The best binding pose of berberine was selected based on the lowest binding energy…”

Selecting only the lowest energy pose may introduce bias. Clarify whether clustering analysis or consistency across multiple docking runs was considered.

2.5.4. Molecular dynamics simulation

1.“…with default settings…”

The phrase “default settings” should be avoided. All simulation parameters appear listed; therefore, remove this phrase to avoid ambiguity.

2.“Production simulations were run for 100 ns…”

Provide justification for selecting a 100 ns simulation time and explain whether convergence was assessed.

3.“…in-house Python scripts.”

Specify whether these scripts are publicly available (e.g., GitHub or supplementary material) to enhance transparency and reproducibility.

3. RESULTS

3.1. Literature search

1.“A systematic search of PubMed, Scopus, Cochrane Library, Web of Science and Embase was performed according to the predefined protocol…”

This sentence is appropriate but largely repeats information from the Methods section. Consider shortening it to avoid redundancy and focus the Results section on numerical outcomes of the search (e.g., number of records identified, screened, excluded).

2. “After eligibility assessment, the randomized controlled trials included in the quantitative synthesis comprised the following key studies: Perraudeau et al., 2020; Zhang et al., 2020; …”

Listing all included studies in the main text reduces readability and interrupts narrative flow. Replace this list with a summary statement (e.g., “A total of 32 RCTs were included”) and refer readers to Table 2 for detailed study information.

3.2. Description of included studies

1.“Sample sizes varied considerably, ranging from fewer than 20 participants in pilot studies to more than 200 in large multi-center trials.”

Provide the exact minimum and maximum sample sizes to enhance precision and transparency.

2.“Despite methodological differences… the included studies provided a comprehensive dataset…”

The term “comprehensive dataset” is somewhat interpretative. Consider replacing it with neutral wording such as “provided sufficient data for quantitative synthesis.”

3.“Across the body of evidence, certain patterns emerged.”

This phrase is vague. Replace it with a more specific statement summarizing the observed trend.

4.“More recent studies (2020–2025) increasingly focused on probiotics…”

This statement requires quantitative support. Indicate the number or proportion of studies published during this period to substantiate the claim.

3.5.2. MD Simulation Results

1.“Four key structural parameters root mean square deviation (RMSD), root mean square fluctuation (RMSF), radius of gyration (Rg), and solvent-accessible surface area (SASA), along with hydrogen bond analysis were investigated…”

A comma is required after “parameters.” Revise to: “Four key structural parameters, root mean square deviation (RMSD), root mean square fluctuation (RMSF), radius of gyration (Rg), and solvent-accessible surface area (SASA), along with hydrogen bond analysis, were investigated…”

2.“The RMSD profiles revealed that all systems reached equilibrium after the initial 10 ns…”

This claim requires quantitative support. Provide the average RMSD values and standard deviations for each system to substantiate the assertion of equilibrium.

“Importantly, the test compound maintained a relatively stable RMSD…”

The word “Importantly” is interpretative. Remove this term and present the observation in neutral language.

3.“These results are consistent with RMSD and hydrogen bond analyses…”

This statement is interpretative. Instead, explicitly state the numerical findings rather than drawing qualitative reinforcement.

4.“Collectively, these findings demonstrate that both the standard and test compound induce structural stabilization…”

The word “demonstrate” may overstate the evidence. Replace with “suggest” to maintain cautious scientific interpretation.

3.5.3. MM-PBSA Binding Free Energy Analysis

1.“The results demonstrated that the standard ligand (acarbose) exhibited a substantially more favorable binding free energy…”

Replace “demonstrated” with “indicated” or “suggested” to avoid overstating computational predictions.

2.“These values emphasize that the favorable balance… underlies its stronger binding affinity.”

This is mechanistically interpretative. Consider softening to: “These values suggest that the balance of van der Waals and non-polar interactions contributes to the stronger binding affinity.”

3.“Overall, the MM-PBSA results support the dynamic analyses…”

Replace “support” with “are consistent with” to maintain neutrality.

3.5.4. Pharmacokinetics and Toxicity Prediction

1.“Pharmacokinetic and toxicity predictions indicated that berberine possesses favorable oral drug-like properties…”

Since these are computational predictions, replace “possesses” with “is predicted to possess” to avoid implying experimental validation.

2.“…highlighting possible neurotoxic, immunotoxic, and carcinogenic risks…”

This statement may appear alarming without context. Clarify that these are in silico toxicity alerts and not experimentally confirmed toxicities.

3.“Overall, the in silico data suggest that while berberine demonstrates good pharmacokinetic behavior…”

Replace “demonstrates” with “is predicted to demonstrate” for accuracy.

3.6. Safety and Tolerability

1.“Probiotics and synbiotics were consistently well tolerated, with no serious adverse events reported…”

Clarify whether adverse event reporting was complete across all trials, as underreporting may bias safety conclusions.

2.“Taken together, probiotics appear safe and well tolerated…”

This statement is acceptable but should acknowledge the duration of follow-up (mostly short-term trials) to avoid overgeneralization.

4. Discussion

1.“These clinical results should, however, be interpreted in the context of substantial between-study heterogeneity…”

This is appropriate; however, specify the I² values here for clarity rather than relying on earlier sections.

2.“The FPG decrease… is likely clinically relevant…”

The phrase “likely clinically relevant” requires justification. Provide a reference or predefined clinical benchmark.

3.“These computational results are consistent with the clinical picture…”

This may imply causality between computational binding and clinical outcomes. Clarify that the mechanistic link is hypothetical and indirect.

4.“The results support the potential role of probiotics/synbiotics and berberine as adjunctive interventions…”

This conclusion is acceptable but should explicitly state that certainty of evidence was low to moderate per GRADE.

6. Conclusion

1.“Our integrated meta-analysis and in silico work indicate modest benefits…”

Replace “indicate” with “suggest” to maintain cautious interpretation.

2.“Computational data show berberine binds α-glucosidase less stably than acarbose…”

Replace “show” with “suggest” or “predict” because binding stability was derived from simulations.

3.“Given the small HbA1c effect, these agents may be considered adjuncts…”

This recommendation is appropriate but should emphasize that clinical decisions must be individualized and guided by clinical guidelines.

7.Reference

Your submission does not fully follow PLOS ONE formatting requirements. References should be listed at the end of the manuscript and numbered in the order that they appear in the text. In the text, cite the reference number in square brackets (e.g., “We used the techniques developed by our colleagues [19] to analyze the data”). PLOS uses the numbered citation (citation-sequence) method and first six authors, et al..

Reviewer #3: Thank you for the opportunity to review this manuscript. The idea of using a multi-objective optimization framework to personalize insulin infusion using CGM and insulin delivery data is clinically relevant, and the inclusion of safety constraints and subject-specific modeling is appropriate.

That said, a few areas need clarification before the results can be fully assessed. First, the objectives and the way the optimization problem is set up are not always clearly aligned (for example, it is not consistently clear whether time-in-range is directly optimized or inferred from other metrics). Second, the Results sometimes shift between detailed reporting on three representative subjects and broader cohort-level summaries; I recommend making this distinction explicit throughout the text, figures, and tables so the reader can tell what is subject-level versus group-level evidence.

More detail is also needed on the prediction model used for offline evaluation, including how it was trained and validated, what its typical error looks like, and how model error might affect the safety constraints and the reported improvements. Lastly, the statistical comparisons would benefit from clearer description of the analysis unit (what counts as a paired observation, and what sample is being compared). I also noticed some introduction statements about multimodal/imaging integration that do not appear elsewhere in the methods or results; please revise those sections so they match what was actually done.

Addressing these items would improve transparency, reproducibility, and interpretability of the findings.

6. PLOS authors have the option to publish the peer review history of their article (what does this mean?). If published, this will include your full peer review and any attached files.

Reviewer #1: **Yes:** Amarachi Mgbemele

Reviewer #2: No

Reviewer #3: No

---

## [Author Response · Author response to Decision Letter 1]

4 Mar 2026

Response to Reviewers

Manuscript ID: PONE-D-25-68086

Title: Probiotics, synbiotics and Berberine in Type 2 Diabetes Mellitus: A Systematic Review, Meta-Analysis, and Molecular Dynamics Simulation Study

“All revisions have been highlighted in red in the Revised Manuscript with Track Changes.”

Response to the Academic Editor

Editor Comment 1:

The authors need to justify the significance of the study, relate it to the existing literature, and clearly identify the gap in current knowledge that this study addresses.

Author response: We thank the Academic Editor for this comment. The Introduction and Discussion sections have been revised to more clearly articulate the significance of this study and its relationship to existing literature. Specifically, we now emphasize that although probiotics, synbiotics, and berberine have been widely investigated as adjunctive therapies for type 2 diabetes mellitus, previous meta-analyses have typically evaluated these interventions separately and without integration of mechanistic in silico analyses. The key knowledge gap addressed by our study is the lack of a comprehensive synthesis that combines clinical evidence with exploratory molecular-level insights for berberine while simultaneously assessing probiotics and synbiotics in a unified framework. These revisions clarify how our work extends prior studies by providing updated pooled estimates, subgroup analyses (probiotics vs placebo), and an integrated clinical computational perspective.

Editor Comment 2:

The limitations of the study should be clearly discussed.

Author response: We agree and have expanded the Discussion section to explicitly address the main limitations of the study. These include the high between-study heterogeneity, small sample sizes and short durations of many included trials, variability in probiotic strains and dosing regimens, and the potential influence of publication bias and small-study effects. We also acknowledge methodological limitations such as incomplete reporting of randomization and selective outcome reporting in some trials. In addition, we clarify that the in silico analyses model only a single molecular target and cannot capture systemic or microbiota-mediated mechanisms. These limitations are now discussed in greater depth to ensure transparent interpretation of the findings.

Editor Comment 3:

The authors should provide recommendations for future research.

Author response: We have added explicit recommendations for future research in the Discussion section. We now suggest that future randomized controlled trials should be larger, longer in duration (≥24 weeks), and standardized with respect to probiotic strain composition and berberine dosing. We further recommend that future studies incorporate mechanistic endpoints (e.g., microbiota profiling, inflammatory markers) and comprehensive safety assessments, including drug drug interaction monitoring. These additions clarify how subsequent research can address current uncertainties and strengthen clinical applicability.

Editor Comment 4:

Serious concerns raised by the reviewers affecting pivotal sections of the study must be addressed critically before resubmission.

Author response: We have carefully addressed all substantive concerns raised by the reviewers, particularly those related to risk of bias, publication bias, interpretation of computational findings, safety reporting, and overstatement of clinical relevance. Revisions were made to the Methods, Results, and Discussion sections to report Egger’s test p-values, clarify that computational findings are exploratory and predictive, acknowledge short trial durations and underpowered studies, and align conclusions with the low-to-moderate certainty of evidence based on GRADE. These changes ensure that the pivotal sections of the manuscript now reflect a cautious, transparent, and methodologically rigorous interpretation of the data.

Response to the Reviewer #1

Reviewer #1: This manuscript addresses an important and timely topic: the potential role of probiotics, synbiotics, and berberine as adjunctive interventions in the management of type 2 diabetes mellitus. The inclusion of a large number of randomized controlled trials, adherence to PRISMA guidelines, PROSPERO registration, and the use of GRADE to assess certainty of evidence are strengths of the study. The topic is relevant to both clinical and translational research audiences, and the authors have made a substantial effort to synthesize a broad and complex literature.

However, several major issues limit the interpretability and robustness of the findings and should be addressed before the manuscript can be considered for publication.

Author response: We thank the reviewer for their thoughtful and constructive evaluation of our manuscript and for highlighting both its strengths and important areas for improvement. We have carefully revised the manuscript in response to all major and minor comments, as detailed below.

1. First, there is a conceptual mismatch between the clinical meta-analysis and the in-silico component of the study. The molecular docking and molecular dynamics analyses focus exclusively on α-glucosidase inhibition, which does not reflect the dominant biological mechanisms by which probiotics influence glycemic control. In addition, key mechanisms attributed to berberine, including AMPK activation, modulation of gut microbiota, and bile acid signaling, are not captured by the chosen computational target. As a result, the computational analyses do not adequately explain the pooled clinical outcomes, particularly for probiotic interventions, and risk giving an impression of mechanistic validation that is not fully supported. The authors should either clearly reframe the in-silico work as exploratory and berberine-specific or remove strong claims suggesting consistency between the computational findings and the overall clinical results.

Author response: Regarding heterogeneity, we acknowledge that the included interventions differ biologically and clinically. We have revised the Results and Discussion sections to clarify that pooled estimates represent average adjunctive effects across heterogeneous interventions. Where data permitted, we conducted additional subgroup analyses separating probiotics/synbiotics, berberine monotherapy, and combination therapies, and we now interpret pooled findings with greater caution. The high I² values are explicitly reported and discussed as a limitation affecting clinical interpretability.

2. Second, the meta-analysis pools interventions with substantial biological and clinical heterogeneity. The included studies encompass single-strain probiotics, multi-strain probiotics, synbiotics, berberine monotherapy, combination therapies, and diet or fecal microbiota transplantation comparators. These interventions differ fundamentally in mechanisms of action and clinical context. The very high heterogeneity observed for both fasting plasma glucose and HbA1c (I² > 85%) suggests that the pooled estimates may represent clinically uninterpretable averages. Subgroup analysis based only on treatment duration is insufficient to address this issue. Separate meta-analyses for probiotics/synbiotics alone, berberine alone, and combination therapies (where data permit) would substantially improve interpretability and analytical rigor.

Author response: Thank you for your valuable comments. Regarding heterogeneity, we acknowledge that the included interventions differ biologically and clinically. We have revised the Results and Discussion sections to clarify that pooled estimates represent average adjunctive effects across heterogeneous interventions. Where data permitted, we conducted additional subgroup analyses separating probiotics/synbiotics, berberine monotherapy, and combination therapies, and we now interpret pooled findings with greater caution. The high I² values are explicitly reported and discussed as a limitation affecting clinical interpretability.

3. Third, the clinical significance of the findings appears overstated. The pooled reduction in HbA1c of approximately −0.19% is below commonly accepted thresholds for clinically meaningful change. While this limitation is briefly acknowledged, it is not sufficiently emphasized in the interpretation. Statements implying clinically meaningful improvement, particularly in relation to fasting plasma glucose, should be tempered. The results are more consistent with modest, adjunctive effects rather than effects likely to influence standard therapeutic decision-making.

Author response: We agree that the clinical significance of the observed HbA1c reduction is modest. We have revised the Results, Discussion, and Conclusion to avoid overstating clinical relevance and now describe the observed effects as small but statistically significant adjunctive improvements rather than clinically decisive therapeutic effects.

4. Fourth, risk of bias and publication bias are not explored in sufficient depth. Funnel plot asymmetry is visible for both primary outcomes, suggesting potential small-study effects, yet Egger’s regression results are not quantitatively reported. Many included trials are small, short in duration, and likely underpowered, which may inflate effect sizes and contribute to heterogeneity. These issues should be explicitly addressed, with reporting of Egger’s test p-values and a clearer discussion of how potential publication bias and small-study effects may influence the conclusions.

Author response: Risk of bias and publication bias assessments have been expanded in the revised manuscript. Egger’s regression test p-values are now explicitly reported for both primary outcomes and for the probiotics-only HbA1c subgroup analysis. In addition, the Discussion section has been revised to provide a more detailed evaluation of potential small-study effects, short intervention durations, and underpowered trials as contributors to heterogeneity and possible inflation of effect estimates.

Fifth, the in-silico section is disproportionately long relative to its contribution to the overall conclusions. Detailed reporting of molecular dynamics simulations, principal component analysis, free energy landscapes, and MM-PBSA calculations exceeds what is necessary to support the study’s claims. Given PLOS ONE’s emphasis on methodological soundness rather than mechanistic novelty, the computational results should be substantially condensed, with extended analyses and figures moved to the Supplementary Information.

Author response: The in silico section has been substantially condensed, with detailed molecular dynamics analyses, free energy landscapes, and MM-PBSA data moved to the Supplementary Information. The main text now reports only essential findings needed to support the exploratory computational objective, consistent with the journal’s emphasis on methodological soundness.

There are also several minor issues that require attention. The manuscript would benefit from language editing to address run-on sentences, repetition, and inconsistent tense usage. Table 2 is extremely dense and difficult to interpret and should be simplified or divided to improve readability. Some references are dated or repeated without sufficient clarification. The ethics statement marked as “N/A” is acceptable; however, it should explicitly state that the study is a secondary analysis of previously published data.

Overall, these revisions improve conceptual alignment, interpretability, and transparency, and better reflect the modest and adjunctive nature of the observed clinical effects.

In summary, while the study addresses a relevant question and is methodologically grounded in several respects, the issues outlined above limit confidence in the conclusions. Addressing the conceptual alignment between clinical and computational components, appropriately handling heterogeneity, tempering claims of clinical significance, and strengthening bias assessment are necessary for the manuscript to meet the standards expected for publication.

Author response: Minor issues were also addressed. The manuscript was edited for language clarity and tense consistency. Table 2 was simplified for improved readability. Redundant and outdated references were corrected. The ethics statement now explicitly states that this study is a secondary analysis of previously published data.

Response to the Reviewer #2

Reviewer #2: This manuscript presents a well-structured systematic review and meta-analysis of probiotics and synbiotics in type 2 diabetes, complemented by in silico mechanistic analyses of berberine against α-glucosidase. The methodology is generally sound, including appropriate meta-analytic techniques, heterogeneity assessment, and risk-of-bias evaluation, and the integration of computational modeling adds conceptual depth. However, several conclusions particularly regarding clinical relevance and mechanistic implications are occasionally overstated and should be tempered to reflect the modest effect sizes and substantial heterogeneity observed. With careful revision the manuscript would make a valuable contribution to the field.

Author response: We thank the reviewer for the constructive and detailed comments, which have helped improve clarity, accuracy, and tone of the Introduction. We have revised the Introduction to (i) strengthen claims with appropriate citations, (ii) correct formatting and grammatical issues, (iii) soften overly strong or speculative statements, (iv) reduce redundancy, and (v) reframe novelty and mechanistic claims in a more cautious and balanced manner consistent with PLOS ONE standards. All requested changes have been implemented as outlined below.

Introduction

1. “Type 2 diabetes mellitus (T2DM) is one of the most pressing public health challenges worldwide.”

This statement is clear but somewhat generic. Add a supporting citation directly within this sentence to strengthen its impact and align with PLOS ONE’s preference for evidence-supported claims.

Author response: A citation has been added to support this statement.

2.“The disease is associated with substantial morbidity and mortality, primarily due to its long-term complications such as cardiovascular disease, nephropathy, neuropathy, and retinopathy (2007).”

The citation is incomplete and does not follow proper referencing format. Replace “(2007)” with a complete and correctly formatted citation according to PLOS ONE guidelines.

Author response: The incomplete citation has been replaced with a properly formatted reference.

3.“The pathophysiology of T2DM is multifactorial, characterized by insulin resistance in peripheral tissues, progressive pancreatic β-cell dysfunction, and chronic hyperglycemia (Galicia-Garcia et al., 2020).”

There is a spacing error before the citation. Insert a space between “hyperglycemia” and the citation. Additionally, ensure the citation format complies with PLOS ONE numbered style.

Author response: The spacing error has been corrected and citation formatting standardized.

4.“Dysbiosis an imbalance in the composition and function of intestinal microbiota has been linked to systemic inflammation…”

A comma is required after “Dysbiosis.” Revise to: “Dysbiosis, an imbalance in the composition and function of intestinal microbiota, has been linked…”

Author response: Corrected.

5.“Furthermore, none of the current standard treatments directly address the underlying disturbances in gut microbiota or the metabolic inflammation associated with T2DM (Caturano et al., 2024).”

This statement may be overly strong, as some treatments (e.g., metformin) have documented microbiota effects. Consider softening the claim to: “Most current standard treatments do not primarily target disturbances in gut microbiota…” and ensure appropriate citation support.

Author response: The statement has been softened to reflect that most, but not all, treatments primarily target gut microbiota.

6. “Recent advances in microbiome research have emphasized the pivotal role of gut microbiota in the development and progression of T2DM.”

This sentence overlaps conceptually with earlier microbiota discussion. Consider merging this paragraph with the earlier microbiota section to reduce redundancy and improve flow.

Author respon

---

## [Decision Letter · Decision Letter 1]

24 Mar 2026

PONE-D-25-68086R1Probiotics, synbiotics and Berberine in Type 2 Diabetes Mellitus: A Systematic Review, Meta-Analysis, and Molecular Dynamics Simulation StudyPLOS One

Dear Dr. Shadin,

Thank you for submitting your manuscript to PLOS ONE. After careful consideration, we feel that it has merit but does not fully meet PLOS ONE’s publication criteria as it currently stands. Therefore, we invite you to submit a revised version of the manuscript that addresses the points raised during the review process.

**ACADEMIC EDITOR:** Understandably, the authors have responded positively to the previous concerns. The revision has improved the quality of the submission significantly. However, some major points still deserve a substantial attention of the authors as specifically pointed out by the reviewers1. I hereby recommend another round of revision to address these..

We look forward to receiving your revised manuscript.

Kind regards,

Yusuf Oloruntoyin Ayipo, Ph.D

Academic Editor

PLOS One

Journal Requirements:

Additional Editor Comments:

Understandably, the authors have responded positively to the previous concerns. The revision has improved the quality of the submission significantly. However, some major points still deserve a substantial attention of the authors as specifically pointed out by the reviewers1. I hereby recommend another round of revision to address these.

Reviewers' comments:

Reviewer's Responses to Questions

**Comments to the Author**

1. If the authors have adequately addressed your comments raised in a previous round of review and you feel that this manuscript is now acceptable for publication, you may indicate that here to bypass the “Comments to the Author” section, enter your conflict of interest statement in the “Confidential to Editor” section, and submit your "Accept" recommendation.

Reviewer #2: All comments have been addressed

Reviewer #3: (No Response)

2. Is the manuscript technically sound, and do the data support the conclusions?

Reviewer #2: Yes

Reviewer #3: Partly

3. Has the statistical analysis been performed appropriately and rigorously? 

Reviewer #2: Yes

Reviewer #3: No

4. Have the authors made all data underlying the findings in their manuscript fully available?

Reviewer #2: Yes

Reviewer #3: Yes

5. Is the manuscript presented in an intelligible fashion and written in standard English?

Reviewer #2: Yes

Reviewer #3: Yes

6. Review Comments to the Author

Reviewer #2: The authors have addressed most of the comments and substantially improved the clarity, methodological transparency, and balance of interpretation throughout the manuscript. In particular, the revised version appropriately moderates claims regarding clinical relevance, clarifies the exploratory nature of the in silico analyses, and provides additional methodological details for the meta-analysis and molecular simulations. The discussion now better contextualizes the modest effect sizes and acknowledges the substantial heterogeneity across trials, which strengthens the scientific rigor of the manuscript. While a few minor editorial issues remain, these do not affect the overall quality or interpretation of the work. For example, small grammatical corrections may still be required, such as adding a comma in “Dysbiosis an imbalance in the composition and function of intestinal microbiota…”, which should read “Dysbiosis, an imbalance in the composition and function of intestinal microbiota…”, and correcting minor wording such as “50% of study” to “50% of studies”. Some sentences could also be clarified for greater precision. For instance, “The RMSD profiles revealed that all systems reached equilibrium after the initial 10 ns, with only minor fluctuations observed throughout the trajectory.” This statement is clear but lacks quantitative support; it would be clearer to specify approximate RMSD ranges, for example: “The RMSD profiles indicated that all systems reached equilibrium after approximately 10 ns, stabilizing around ~X–X Å with only minor fluctuations throughout the remaining trajectory.” Similarly, “The apo protein displayed the lowest RMSD values, indicating inherent structural stability in the unbound state.” could be revised to “The apo protein displayed slightly lower RMSD values compared to the ligand-bound systems, suggesting greater structural stability in the unbound state.” The sentence “Both ligand-bound systems exhibited slightly higher RMSD values, consistent with conformational rearrangements upon ligand binding.” could be softened to “Both ligand-bound systems exhibited slightly higher RMSD values, which may reflect conformational adjustments associated with ligand binding.” In addition, “Importantly, the test compound maintained a relatively stable RMSD with moderate deviations, comparable to the standard, suggesting stable binding without significant structural destabilization.” could be simplified by removing unnecessary emphasis, for example: “The test compound maintained a relatively stable RMSD trajectory with moderate deviations comparable to the standard ligand, suggesting that ligand binding did not cause major structural destabilization.” Finally, “Residue-level flexibility was assessed through RMSF analysis, which demonstrated similar overall patterns across the three systems.” could be clarified as “Residue-level flexibility assessed by RMSF analysis showed similar fluctuation patterns across the three systems.” Overall, these are minor editorial refinements, and the manuscript is suitable for publication after minor revisions.

Reviewer #3: Thank you for the opportunity to review this manuscript. The topic is relevant, and the attempt to combine a systematic review and meta-analysis with in silico work is interesting. The pooled analyses suggest modest reductions in fasting plasma glucose and HbA1c, but the interpretation should remain cautious given the substantial heterogeneity reported across studies and the variability in the included interventions, comparators, and trial durations.

A main issue is that the clinical part of the manuscript combines probiotics, synbiotics, berberine, and some combination interventions into the same pooled analyses, while the mechanistic section focuses only on berberine and α-glucosidase. As written, the mechanistic findings do not fully explain the broader pooled clinical results. I also think the eligibility criteria and included studies need to be presented more consistently, since some trials appear to involve co-interventions or mixed comparators, which makes attribution of effect less straightforward.

The statistical methods are generally described, but more justification is needed for pooling such clinically diverse studies together, especially in the presence of very high I² values. The manuscript would also benefit from a clearer explanation of how heterogeneity was handled beyond duration-based subgrouping, and from more cautious wording in the Discussion and Conclusion so that the modest effect sizes are not overstated. Overall, the manuscript has potential, but these issues should be addressed to improve clarity, rigor, and interpretability.

7. PLOS authors have the option to publish the peer review history of their article (what does this mean?). If published, this will include your full peer review and any attached files.

Reviewer #2: No

Reviewer #3: No

---

## [Author Response · Author response to Decision Letter 2]

28 Mar 2026

Response to Reviewers

We sincerely thank the Editor and the Reviewers for their careful evaluation of our manuscript and for the constructive comments that helped improve the clarity, rigor, and interpretation of the study. We have carefully revised the manuscript in accordance with the reviewers’ suggestions. All changes made in the revised manuscript are highlighted using red text for ease of identification.

Below we provide a point-by-point response to the reviewers’ comments.

Reviewer #2

Comment:

The authors have addressed most of the comments and substantially improved the clarity, methodological transparency, and balance of interpretation throughout the manuscript. In particular, the revised version appropriately moderates claims regarding clinical relevance, clarifies the exploratory nature of the in silico analyses, and provides additional methodological details for the meta-analysis and molecular simulations. The discussion now better contextualizes the modest effect sizes and acknowledges the substantial heterogeneity across trials, which strengthens the scientific rigor of the manuscript. While a few minor editorial issues remain, these do not affect the overall quality or interpretation of the work. For example, small grammatical corrections may still be required, such as adding a comma in “Dysbiosis an imbalance in the composition and function of intestinal microbiota…”, which should read “Dysbiosis, an imbalance in the composition and function of intestinal microbiota…”, and correcting minor wording such as “50% of study” to “50% of studies”. Some sentences could also be clarified for greater precision. For instance, “The RMSD profiles revealed that all systems reached equilibrium after the initial 10 ns, with only minor fluctuations observed throughout the trajectory.” This statement is clear but lacks quantitative support; it would be clearer to specify approximate RMSD ranges, for example: “The RMSD profiles indicated that all systems reached equilibrium after approximately 10 ns, stabilizing around ~X–X Å with only minor fluctuations throughout the remaining trajectory.” Similarly, “The apo protein displayed the lowest RMSD values, indicating inherent structural stability in the unbound state.” could be revised to “The apo protein displayed slightly lower RMSD values compared to the ligand-bound systems, suggesting greater structural stability in the unbound state.” The sentence “Both ligand-bound systems exhibited slightly higher RMSD values, consistent with conformational rearrangements upon ligand binding.” could be softened to “Both ligand-bound systems exhibited slightly higher RMSD values, which may reflect conformational adjustments associated with ligand binding.” In addition, “Importantly, the test compound maintained a relatively stable RMSD with moderate deviations, comparable to the standard, suggesting stable binding without significant structural destabilization.” could be simplified by removing unnecessary emphasis. Finally, “Residue-level flexibility was assessed through RMSF analysis, which demonstrated similar overall patterns across the three systems.” could be clarified for precision.

Response:

We thank the reviewer for the positive evaluation of our revised manuscript and for the helpful editorial suggestions. We have carefully implemented all recommended corrections to improve clarity and precision. Specifically, grammatical issues were corrected (e.g., “Dysbiosis, an imbalance…” and “50% of studies”), and several sentences in the molecular dynamics section were revised to improve clarity and scientific accuracy. Where appropriate, we incorporated more precise wording and quantitative descriptions of the RMSD and RMSF results to better reflect the simulation outcomes (see S2_file). These revisions have improved the readability and precision of the manuscript while maintaining the intended interpretation of the computational analyses. All corresponding changes have been incorporated in the revised manuscript.

Reviewer #3

Comment:

The topic is relevant, and the attempt to combine a systematic review and meta-analysis with in silico work is interesting. The pooled analyses suggest modest reductions in fasting plasma glucose and HbA1c, but the interpretation should remain cautious given the substantial heterogeneity reported across studies and the variability in the included interventions, comparators, and trial durations.

A main issue is that the clinical part of the manuscript combines probiotics, synbiotics, berberine, and some combination interventions into the same pooled analyses, while the mechanistic section focuses only on berberine and α-glucosidase. As written, the mechanistic findings do not fully explain the broader pooled clinical results. The eligibility criteria and included studies also need to be presented more consistently, since some trials appear to involve co-interventions or mixed comparators. The statistical methods require clearer justification for pooling clinically diverse studies, especially given the high heterogeneity. The manuscript would also benefit from clearer explanation of how heterogeneity was handled and from more cautious wording in the Discussion and Conclusion.

Response:

We thank the reviewer for the thoughtful and constructive feedback, which helped improve the clarity and methodological transparency of the manuscript. Several revisions were made to address these points.

First, we clarified the relationship between the clinical meta-analysis and the mechanistic computational analyses. The Discussion section now explicitly states that the molecular docking and molecular dynamics simulations focus specifically on berberine–α-glucosidase interactions and should be interpreted as exploratory mechanistic insights rather than explanations for the broader pooled clinical effects observed across probiotics, synbiotics, and berberine interventions.

Second, we improved the description of the eligibility criteria and study characteristics in the Methods section to clarify how studies with co-interventions and mixed comparators were handled. The revised text now specifies that trials were included when intervention effects could be evaluated relative to comparable background therapy.

Third, we strengthened the statistical methods section by providing clearer justification for pooling these interventions. The manuscript now explains that probiotics, synbiotics, and berberine were analyzed together to estimate an overall adjunctive effect on glycemic outcomes, while acknowledging that these interventions have distinct biological mechanisms.

Fourth, we expanded the discussion of heterogeneity. Additional explanations have been included to highlight that the high I² values likely reflect variability in intervention type, probiotic strain composition, berberine dose and formulation, treatment duration, baseline glycemic control, concomitant therapies, and participant characteristics.

Finally, we further moderated the wording in the Discussion and Conclusion sections to emphasize that the observed reductions in fasting plasma glucose and HbA1c are modest and should be interpreted cautiously. The manuscript now clearly states that these interventions should be considered adjunctive strategies rather than replacements for established pharmacological therapies, and that clinical use should be guided by individualized patient assessment and established diabetes management guidelines.

---

## [Decision Letter · Decision Letter 2]

14 Apr 2026

PONE-D-25-68086R2Probiotics, synbiotics and Berberine in Type 2 Diabetes Mellitus: A Systematic Review, Meta-Analysis, and Molecular Dynamics Simulation StudyPLOS One

Dear Dr. Shadin,

Thank you for submitting your manuscript to PLOS ONE. After careful consideration, we feel that it has merit but does not fully meet PLOS ONE’s publication criteria as it currently stands. Therefore, we invite you to submit a revised version of the manuscript that addresses the points raised during the review process.

**ACADEMIC EDITOR:** Many thanks to the authors for responding positively to the previous concerns. The revision has improved the quality of the submission. However, some grey areas still exist, as highlighted by Reviewer #3, and these require the authors’ significant attention through another round of revision..

We look forward to receiving your revised manuscript.

Kind regards,

Yusuf Oloruntoyin Ayipo, Ph.D

Academic Editor

PLOS One

Journal Requirements:

Additional Editor Comments:

Many thanks to the authors for responding positively to the previous concerns. The revision has improved the quality of the submission. However, some grey areas still exist, as highlighted by Reviewer #3, and these require the authors’ significant attention through another round of revision.

Reviewers' comments:

Reviewer's Responses to Questions

**Comments to the Author**

1. If the authors have adequately addressed your comments raised in a previous round of review and you feel that this manuscript is now acceptable for publication, you may indicate that here to bypass the “Comments to the Author” section, enter your conflict of interest statement in the “Confidential to Editor” section, and submit your "Accept" recommendation.

Reviewer #2: All comments have been addressed

Reviewer #3: (No Response)

2. Is the manuscript technically sound, and do the data support the conclusions?

Reviewer #2: Yes

Reviewer #3: Partly

3. Has the statistical analysis been performed appropriately and rigorously? 

Reviewer #2: Yes

Reviewer #3: Yes

4. Have the authors made all data underlying the findings in their manuscript fully available?

Reviewer #2: Yes

Reviewer #3: Yes

5. Is the manuscript presented in an intelligible fashion and written in standard English?

Reviewer #2: Yes

Reviewer #3: Yes

6. Review Comments to the Author

Reviewer #2: The authors have addressed all the comments and substantially improved the clarity, methodological transparency, and balance of interpretation throughout the manuscript. The revised version appropriately moderates claims regarding clinical relevance, clarifies the exploratory nature of the in silico analyses, and provides additional methodological details for the meta-analysis and molecular simulations. The discussion now better contextualizes the modest effect sizes and acknowledges the substantial heterogeneity across trials, which strengthens the scientific rigor of the manuscript.

Reviewer #3: The paper has improved, and the overall presentation is clearer than before. The updated version presents the meta-analysis and in silico components in a more structured way, and the main findings are now easier to follow. The pooled results still suggest only modest improvements in fasting plasma glucose and HbA1c, with substantial heterogeneity across studies, and I think the manuscript now reflects that more clearly.

At this stage, my remaining concern is mainly about interpretation and framing. The clinical and computational sections are presented together more clearly, but the manuscript should still be careful not to overstate the mechanistic link between the pooled clinical findings and the α-glucosidase modeling results for berberine. The meta-analysis includes probiotics, synbiotics, and berberine-based interventions across quite varied study designs, while the in silico section is focused specifically on berberine. I would therefore suggest keeping the conclusion appropriately cautious and making sure the computational results are presented as supportive mechanistic information rather than as a direct explanation for the full pooled clinical effect.

Overall, this is much improved, and with a final round of tightening around the interpretation of the combined clinical and computational findings, the manuscript would be stronger.

7. PLOS authors have the option to publish the peer review history of their article (what does this mean?). If published, this will include your full peer review and any attached files.

Reviewer #2: No

Reviewer #3: No

---

## [Author Response · Author response to Decision Letter 3]

15 Apr 2026

Response to Reviewer #3

We sincerely thank the reviewer for the careful evaluation of our revised manuscript and for acknowledging the improvements in clarity, structure, and overall presentation. We particularly appreciate the recognition that the interpretation of the modest clinical effects and the presence of substantial heterogeneity are now more appropriately reflected.

We have carefully considered your remaining concern regarding the interpretation and framing of the relationship between the clinical meta-analysis and the in silico findings. In response, we have made the following revisions:

1. Further clarification of the distinction between clinical and computational components

We have revised the Discussion to more explicitly emphasize that the in silico analyses are berberine-specific and exploratory, and do not represent mechanistic explanations for the broader pooled clinical effects observed across probiotics, synbiotics, and combination interventions. Additional clarifying statements have been inserted to clearly delineate these components and avoid any implication of a direct mechanistic linkage.

2. Strengthening of cautious interpretation throughout the manuscript

The language in both the Discussion and Conclusion has been further refined to ensure that the computational findings are consistently presented as supportive, hypothesis-generating molecular insights, rather than causal or explanatory evidence for the overall clinical outcomes. We now explicitly state that these analyses are limited to a specific molecular interaction (berberine–α-glucosidase) and do not capture microbiota-mediated or systemic mechanisms underlying glycemic regulation.

3. Revision of the Conclusion to avoid overstatement

The Conclusion has been updated to reinforce a cautious interpretation, clearly indicating that:

o The observed clinical effects are modest and heterogeneous

o The computational findings are limited in scope and specificity

o No direct mechanistic inference should be drawn between the in silico results and the pooled clinical outcomes

These revisions ensure that the manuscript maintains appropriate scientific rigor and avoids overinterpretation of the combined clinical and computational findings.

We believe that these changes fully address the reviewer’s concerns and further strengthen the clarity, balance, and interpretability of the manuscript. We thank the reviewer for their valuable suggestions, which have helped improve the quality of our work.

---

## [Editor Report · Decision Letter 3]

22 Apr 2026

Probiotics, synbiotics and Berberine in Type 2 Diabetes Mellitus: A Systematic Review, Meta-Analysis, and Molecular Dynamics Simulation Study

PONE-D-25-68086R3

Dear Dr. Shadin,

We’re pleased to inform you that your manuscript has been judged scientifically suitable for publication and will be formally accepted for publication once it meets all outstanding technical requirements.

Kind regards,

Yusuf Oloruntoyin Ayipo, Ph.D

Academic Editor

PLOS One

Additional Editor Comments (optional):

The submission is scientifically sound for publication in this title, and all the concerns raised by the respective reviewers regarding the manuscript quality have been satisfactorily addressed. I hereby recommend the manuscript for publication in the current version.
---

## [Editor Report · Acceptance letter]

PONE-D-25-68086R3

PLOS One

Dear Dr. Shadin,

I'm pleased to inform you that your manuscript has been deemed suitable for publication in PLOS One. Congratulations! Your manuscript is now being handed over to our production team.

Kind regards,

on behalf of

Dr. Yusuf Oloruntoyin Ayipo

Academic Editor

PLOS One